# SAFE NEUROSYMBOLIC LEARNING WITH DIFFERENTIABLE SYMBOLIC EXECUTION

**Chenxi Yang, Swarat Chaudhuri**
The University of Texas at Austin

## ABSTRACT

We study the problem of learning worst-case-safe parameters for programs that use neural networks as well as symbolic, human-written code. Such *neurosymbolic* programs arise in many safety-critical domains. However, because they can use nondifferentiable operations, it is hard to learn their parameters using existing gradient-based approaches to safe learning. Our approach to this problem, *Differentiable Symbolic Execution* (DSE), samples control flow paths in a program, symbolically constructs worst-case "safety losses" along these paths, and backpropagates the gradients of these losses through program operations using a generalization of the REINFORCE estimator. We evaluate the method on a mix of synthetic tasks and real-world benchmarks. Our experiments show that DSE significantly outperforms the state-of-the-art DIFFAI method on these tasks.[1]

## 1 INTRODUCTION

Safety on worst-case inputs has recently emerged as a key challenge in deep learning research. Formal verification of neural networks (Albarghouthi, 2021) is an established response to this challenge. In particular, a body of recent work (Mirman et al., 2018; Madry et al., 2017; Cohen et al., 2019; Singh et al., 2018) seeks to incorporate formal verification into the training of neural networks. DIFFAI, among the most prominent of such approaches, uses a neural network verifier to construct a differentiable, worst-case safety loss for the learner. This loss is used to regularize a standard data-driven loss, biasing the learner towards parameters that are both performant and safe.

A weakness of these methods is that they only consider functional properties (such as adversarial robustness) of isolated neural networks. By contrast, in real-world applications, neural networks are often embedded within human-written symbolic code, and correctness requirements apply to the *neurosymbolic programs* (Chaudhuri et al., 2021) obtained by composing the two. For example, consider a car directed by a neural controller (Qin & Badgwell, 2000). Safety properties for the car are functions of its trajectories, and these trajectories depend not just on the controller but also the symbolic equations that define the environment. While recent work (Christakis et al., 2021) has studied the verification of such neurosymbolic programs, there is no prior work on integrating verification and learning for such systems.

In this paper, we present the first steps towards such an integration. The fundamental technical challenge here is that while a neural network is differentiable in its parameters, the code surrounding it may be non-differentiable or even discontinuous. This makes our problem a poor fit for methods like DIFFAI, which were designed for fully differentiable learning objectives.

We solve the problem using a new method, *Differentiable Symbolic Execution* (DSE), that can estimate gradients of worst-case safety losses of nondifferentiable neurosymbolic programs. Our method approximates the program's worst-case safety loss by an integral over the safety losses along individual control flow paths in the program. We compute the gradients of this integral using a generalization of the classic REINFORCE estimator (Williams, 1992). The gradient updates resulting from this process balance two goals: (i) choosing the parameters of individual paths so that program trajectories that follow them are safer, and (ii) learning the parameters of conditional branches so that the program is discouraged from entering unsafe paths. The procedure requires a

---

[1]Our implementation of DSE is available at https://github.com/cxyang1997/DSE.

```
1    Example(x): // x ∈ [−5, 5]
2        y := NN_θ(x)
3        if y ≤ 1.0:
4            z := x + 10.0
5        else :
6            z := x − 5.0
7        assert (z <= 1)
```

$Loc = \{\ell_2, \ell_3, \ell_7\}, X = \{x, y, z\}, l_0 = \{\ell_2\}$
$Init = (-5 \le x \le 5)$
$Safe(l_7) = (z \le 1), Safe(l_2) = \textbf{True}, Safe(l_3) = \textbf{True}$
$Trans_\theta = \{(\ell_2, \textbf{True}, \langle y := NN_\theta(x) \rangle, \ell_3),$
$\qquad\qquad (\ell_3, (y \le 1.0), \langle z := x + 10.0 \rangle, \ell_7),$
$\qquad\qquad (\ell_3, (y > 1.0), \langle z := x - 5.0 \rangle, \ell_7)$

Figure 1: (Left) An example program. $NN_\theta$ is a neural network with parameters $\theta$. DIFFAI fails to learn safe parameters for this program. (Right) The program as an STS. Each location $\ell_i$ corresponds to line $i$ in the program. For brevity, the updates only depict the variable that changes value.

method for sampling paths and computing the safety loss for individual paths. We use a version of *symbolic execution* (Baldoni et al., 2018) to this end.

We evaluate DSE through several problems from the embedded control and navigation domains, as well as several synthetic tasks. Our baselines include an extended version of DIFFAI, the current state of the art, and an ablation that does not use an explicit safety loss. Our experiments show that DSE significantly outperforms the baselines in finding safe and high-performance model parameters.

To summarize, our main contributions are:

- We present the first approach to worst-case-safe parameter learning for neural networks embedded within nondifferentiable, symbolic programs.

- As part of our learning algorithm, we give a new way to bring together symbolic execution and stochastic gradient estimators. that might have applications outside our immediate task.

- We present experiments that show that DSE can handle programs with complex structure and also outperforms DIFFAI on our problem.

## 2 PROBLEM FORMULATION

**Programs.** We define programs with embedded neural networks as *symbolic transition systems* (STS) (Manna & Pnueli, 2012). Formally, a program $F_\theta$ is a tuple $(Loc, X, l_0, Init, Safe, Trans_\theta)$. Here, $Loc$ is a finite set of *(control) locations*, $X = \{x_1, \ldots, x_m\}$ is a set of real-valued variables, and $l_0 \in Loc$ is the *initial location*. $Init$, a constraint over $X$, is an initial condition for the program. $Safe$ is a map from locations to constraints over $X$; intuitively, $Safe(l)$ is a safety requirement asserted at location $l$. Finally, $Trans_\theta$ is a *transition relation* consisting of *transitions* $(l, G, U_\theta, l')$ such that: (i) $l$ is the *source location* and $l'$ is the *destination location*; (ii) the *guard* $G$ is a constraint over $X$; and (iii) the *update* $U_\theta$ is a vector $\langle U_{1,\theta}, \ldots, U_{m,\theta} \rangle$, where each $U_{i,\theta}$ is a real-valued expression over $X$ constructed using standard symbolic operators and neural networks with parameters $\theta$. Intuitively, $U_{i,\theta}$ represents the update to the $i$-th variable. We assume that each $U_{i,\theta}$ is differentiable in $\theta$. We also assume that each guard $G$ is of the form $x_i \bowtie 0$, where $\bowtie \in \{<, >, \le, \ge\}$ — note that this is not a significant restriction as $x_i$ can be made to store the value of a complex expression using an update. Finally, we assume that programs are *deterministic*. That is, if $G$ and $G'$ are guards for two distinct transitions from the same source state, then $G \wedge G'$ is unsatisfiable.

Programs in higher-level languages can be translated to the STS notation in a standard way. For example, Figure 1 (left) shows a simple high-level program. The STS for this program appears in Figure 1 (right). While the program is simple, the state-of-the-art DIFFAI approach to verified learning fails to learn safe parameters for it.

**Safety Semantics.** In classical formal methods, a program is considered safe if all of its executions satisfy a safety constraint. However, in learning settings, it helps to know not only whether a program is unsafe, but also the *extent* to which it is unsafe. Consequently, we define the safety semantics of programs in terms of a *(worst-case) safety loss* that quantifies a program's safeness.

Formally, let a *state* of $F_\theta$ be a pair $s = (l, v)$, where $l$ is a location and $v \in \mathbb{R}^m$ is an *assignment* of values to the variables (i.e., $v(i)$ is the value of $x_i$). Such a state is said to be *at* location $l$. For boolean constraints $B$ and assignments $v$ to variables in $B$, let us write $B(v)$ if $v$ satisfies $B$. A state $(l_0, v)$, where $Init(v)$, is an *initial state*. A state $(l, v)$ is *safe* if $(Safe(l))(v)$.

Let $v \in \mathbb{R}^m$ be an assignment to the variables. For a real-valued expression $E$ over $X$, let $E(v)$ be the value of $E$ when $x_i$ is substituted by $v(i)$. For an update $U = \langle U_1, \ldots, U_n \rangle$, we define $U(v)$

as the assignment $\langle U_1(v), \ldots, U_n(v) \rangle$. A length-$n$ *trajectory* of $F_\theta$ is a sequence $\tau = \langle s_0, \ldots, s_n \rangle$, with $s_i = (l_i, v_i)$, such that: (i) $s_0$ is an initial state; and (ii) for each $i$, there is a transition $(l_i, G, U, l_{i+1})$ such that $G(v_i)$ and $v_{i+1} = U(v_i)$.

Let us fix a size bound $N$ for trajectories. A trajectory $\tau = \langle s_0, \ldots, s_n \rangle$ is *maximal* if it has length $N$, or if there is no trajectory $\tau' = \langle s_0, \ldots, s_n, s_{n+1} \rangle$ with length $\leq N$. Because our programs are deterministic, there is a unique maximal trajectory from each $s_0$. We denote this trajectory by $\tau(s_0)$.

Let us assume a real-valued loss $Unsafe(s)$ that quantifies the *unsafeness* of each state $s$. We require $Unsafe(s) = 0$ if $s$ is safe and $Unsafe(s) > 0$ otherwise. We lift this measure to trajectories $\tau$ by letting $Unsafe(\tau) = \sum_{s \text{ appears in } \tau} Unsafe(s)$. The *safety loss* $C(\theta)$ for $F_\theta$ is now defined as:

$$C(\theta) = \max_{s \text{ is an initial state}} Unsafe(\tau(s)). \tag{1}$$

Thus, $C(\theta) = 0$ if and only if all program trajectories are safe.

**Problem Statement.** Our learning problem formalizes a setting in which we have training data for neural networks inside a program $F_\theta$. While training the networks with respect to this data, we must ensure that the overall program satisfies its safety requirements.

To ensure that the parameters of the different neural networks in $F_\theta$ are not incorrectly entangled, we assume in this exposition that only one of these networks, $\mathbf{NN}_\theta$, has trainable parameters.[2] We expect as input a training set of i.i.d. samples from an unknown distribution over the inputs and outputs of $\mathbf{NN}_\theta$, and a differentiable *data loss* $Q(\theta)$ that quantifies the network's fidelity to this training set. Our learning goal is to solve the following constrained optimization problem:

$$\min_\theta Q(\theta) \qquad \text{s.t. } C(\theta) \leq 0. \tag{2}$$

## 3 Learning Algorithm

The learning approach in DSE is based on two ideas. First, we directly apply a recently-developed equivalence between constrained and regularized learning (Agarwal et al., 2018; Le et al., 2019) to reduce Equation (2) to a series of unconstrained optimization problems. Second, we use the novel technique of *differentiating through a symbolic executor* to solve these unconstrained problems.

At the highest level, we *convexify* the class of programs $F_\theta$ by allowing stochastic combinations of programs. That is, we now allow programs of the form $F_{\hat{\theta}} = \sum_{t=1}^T \alpha_t F_{\theta_t}$, where $\sum_i \alpha_t = 1$. To execute the program $F_\theta$ from a given initial state, we sample a specific program $F_{\theta_t}$ from the distribution $(\alpha_1, \ldots, \alpha_t)$ and then execute it from that state.

Equation (2) can now be written into the problem $\max_{\lambda \in \mathbb{R}^+} \min_{\hat{\theta}} Q(\hat{\theta}) + \lambda C(\hat{\theta})$, which in turn can be solved using a classic algorithm (Freund & Schapire, 1999) for computing equilibria in a two-player game. See Appendix A.1 for more details on this algorithm.

A key feature of our high-level algorithm is that it repeatedly solves the optimization problem $\min_\theta Q(\theta) + \lambda C(\theta)$ for fixed values of $\lambda$. This problem is challenging because while $Q(\theta)$ is differentiable in $\theta$, $C(\theta)$ depends on the entirety of $F_\theta$ and may not even be continuous. DSE, our main contribution, addresses this challenge by constructing a *differentiable approximation* $C^\#(\theta)$ of $C(\theta)$. We present the details of this method in the next section.

## 4 Differentiating through a Symbolic Executor

**Background on Symbolic Execution.** DSE uses a refinement of *symbolic execution* (Baldoni et al., 2018), a classic technique for systematic formal analysis of programs. Symbolic execution has been recently used to analyze the safety of neural networks (Gopinath et al., 2018). However, we are not aware of any prior attempt to incorporate symbolic execution into neural network training.

---

[2]Note that some of our experimental benchmarks have $k > 1$ trainable neural modules. In these tasks, each neural module comes with its own training data; thus, in effect, we have $k$ instances of the learning problem. Our implementation learns the modules simultaneously by interleaving their gradient updates. However, the gradient of $Q$ for each module is only influenced by its own data.

A symbolic executor systematically searches the set of *symbolic trajectories* of programs, which we now define. Consider a program $F_\theta = (Loc, X = \{x_1, \ldots, x_m\}, l_0, Init, Safe, Trans_\theta)$ as in Section 2. First, we fix a syntactically restricted class $\mathcal{V}_\theta$ of boolean constraints over $X$ and $\theta$. $\mathcal{V}_\theta$ is required to be closed under conjunction and must include all expressions that can appear as a guard in $F_\theta$. In particular, in our implementation, $\mathcal{V}_\theta$ consists of formulas that specify closed or open intervals, with bounds given by expressions over parameters, in which each variable lies.

A *symbolic state* of $F_\theta$ is a pair $\sigma_\theta = (l, V_\theta)$, where $l$ is a location and $V_\theta \in \mathcal{V}_\theta$. Intuitively, $\sigma_\theta$ represents the set of states of $F_\theta$ that are at location $l$ and satisfy the property $V_\theta$. We designate a finite set of *initial symbolic states* $(l_0, V_0^1), \ldots, (l_p, V_0^p)$. It is required that $Init \Rightarrow \bigvee_i V_0^i$. Intuitively, the initial symbolic states "cover" all the possible initial states of the program $F_\theta$.

We also construct an *overapproximate update* $U_\theta^\# : \mathcal{V}_\theta \to \mathcal{V}_\theta$ for each update $U_\theta$ in $F_\theta$. For all $V \in \mathcal{V}_\theta$, we have $Post(v) \Rightarrow U_\theta^\#(V)$, where $Post(v)$ is the formula $\exists v' : V(v') \wedge (U(v') = v)$. Intuitively, $U_\theta^\#(V)$ is an overapproximate representation of the set of states reached by applying $U_\theta$ to a state that satisfies $V$.

Let us call a transition $t = (l, G, U, l')$ *enabled* at a symbolic state $(l, V)$ if $(G \wedge V)$ is satisfiable. A *symbolic trajectory* of $F_\theta$ is a sequence $\tau_\theta^\# = \langle \sigma_0, \ldots, \sigma_n \rangle$ with the following properties:

- $\sigma_0$ is an initial symbolic state.
- Let $\sigma_i = (l_i, V_i)$ and $\sigma_{i+1} = (l_{i+1}, V_{i+1})$. Then there is a transition $t = (l_i, G_i, U_i, l_{i+1})$ such that: (i) $t$ is enabled at $\sigma_i$, and (ii) $V_{i+1} = U^\#(G_i \wedge V_i)$. In this case, we write $\sigma_{i+1} = t(\sigma_i)$.

Intuitively, $\tau_\theta^\#$ represents an overapproximation of the set of concrete trajectories of $F_\theta$ that pass through the "control path" $\langle l_0, \ldots, l_n \rangle$.

The above symbolic trajectory is *safe* if for all $i$, $V_i \Rightarrow Safe(l_i)$. Intuitively, in this case, all concrete trajectories that the symbolic trajectory represents follow the program's safety requirements.

**Example(Cont.)** Consider the example program in Figure 1. Let us assume a single initial symbolic state where $x \in [-5, 5]$, and suppose $\mathbf{NN}_\theta(x) \in [-2, 2]$ when $x \in [-5, 5]$. The program has two symbolic trajectories from the initial symbolic state:

- $\tau_1^\# = \langle (\ell_2, x \in [-5, 5]), (\ell_3, x \in [-5, 5] \wedge y \in [-2, 2]), (\ell_7, x \in [-5, 5] \wedge y \in [-2, 1] \wedge z \in [5, 15]) \rangle$
- $\tau_2^\# = \langle (\ell_2, x \in [-5, 5]), (\ell_3, x \in [-5, 5] \wedge y \in [-2, 2]), (\ell_7, x \in [-5, 5] \wedge y \in (1, 2] \wedge z \in [-10, 0]) \rangle$.

We note that only $\tau_2^\#$ is safe.

**Probabilistic Symbolic Execution.** A key difficulty with using symbolic execution to estimate the safety loss $C(\theta)$ is that our programs need not be differentiable, or even continuous. To understand the issues here, consider our running example once again. Here, the parameter $\theta$ is used to calculate the variable $y$, which is used in a discontinuous conditional statement that assigns very different values to the variable $z$ in its two branches. Additionally, a slight change in $\theta$ can sometimes cause the guard $G$ of the conditional to go from being **True** on *some* values of $y$, to being **False** on *all* values of $y$. In the second case, the updated $\theta$ would have one, rather than two, symbolic trajectories. If the symbolic trajectory $\tau^\#$ in which $G$ is **True** serves as an edge case to judge whether a program is worst-case-safe, learning safe parameters would be difficult, as there would be no gradient guiding the optimizer back to the case in which $G$ is **True**.

DSE overcomes these difficulties using a probabilistic approach to symbolic execution. Specifically, at a symbolic state $\sigma_i = (l_i, V_i)$, the symbolic executor in DSE *samples* its next action following a probability distribution $p_\theta(t \mid \sigma_i)$, where $t$ ranges over program transitions or a special action $Stop$.

For a boolean expression $V_\theta$ over the program variables $\langle x_0, x_1, \ldots, x_k \rangle$ and parameters $\theta$, let $\mathbf{Vol}(V_\theta)$ denote the *volume* of the assignments to $X$ that satisfy $V_\theta$. We assume a procedure to compute $\mathbf{Vol}(V_\theta)$ for any $V_\theta$. (Note that such a procedure is easy to implement when every $V_\theta$ is an interval, as is the case in our implementation.) We define:

- If there is a unique program transition $t$ that is enabled at $\sigma_i$, then $p_\theta(t|\sigma_i) = 1$.

- If there is no transition $t$ that is enabled at $\sigma_i$, then $p_\theta(Stop|\sigma_i) = 1$.
- Otherwise, let $t_1, \ldots, t_k$ be the transitions that are enabled at $\sigma_i$, with $t_i = (l_i, G_i, U_i, l_{i+1})$. Then $p_\theta(t_j|\sigma_i) = \frac{\mathbf{Vol}(G_j \wedge V_i)}{\mathbf{Vol}(V_i)}$.

We also define a volume-weighted probability distribution $p(\sigma_0)$ over the (finite set of) initial symbolic states $\sigma_0^i$. Specifically, for each $\sigma_0^i = (l_0, V_0^i)$, we have

$$p(\sigma_0^i) = \frac{\mathbf{Vol}(V_0^i)}{\sum_j \mathbf{Vol}(V_0^j)}. \tag{3}$$

DSE uses the distributions $p_\theta(t|\sigma_i)$ and $p(\sigma_0)$ to sample symbolic trajectories. Let $\tau_\theta^\# = \langle \sigma_0, \ldots, \sigma_n \rangle$, with $\sigma_{i+1} = t_i(\sigma_i)$. The probability of sampling $\tau^\#$ is

$$p_\theta(\tau_\theta^\#) = p(\sigma_0) \prod_i p_\theta(t_i|\sigma_{i-1}). \tag{4}$$

We note that $p_\theta$ is differentiable in $\theta$.

**Approximate Safety Loss.** A key decision in DSE is to approximate the overall worst-case loss $C(\theta)$ by the *expectation* of a differentiable safety loss computed per sampled symbolic trajectory. Recall the safety loss $Unsafe(s)$ for individual states. For $\sigma = (l, V)$, we define $Unsafe_\theta(\sigma)$ as a differentiable approximation of the worst-case loss $\max_{s \text{ satisfies } V} Unsafe(s)$. We lift this loss to symbolic trajectories $\tau_\theta^\# = \langle \sigma_0, \ldots, \sigma_n \rangle$ by defining $Unsafe_\theta(\tau^\#) = \sum_i Unsafe_\theta(\sigma_i)$. We observe that $Unsafe_\theta$ is differentiable in $\theta$.

Our approximation to the safety loss is now given by:

$$C^\#(\theta) = \mathbf{E}_{\tau^\# \sim p_\theta(\tau^\#)} Unsafe_\theta(\tau^\#). \tag{5}$$

Intuitively, $C^\#(\theta)$ guides the learner towards either making the unsafe trajectories safer or lowering the probability of the program falling into unsafe trajectories. If $C^\#(\theta)$ equals zero, the program is provably safe. That said, in practice, we must estimate the expectation in Equation (5) using sampling. As this sampling process may miss trajectories, a low empirical estimate of $C^\#(\theta)$ need not guarantee worst-case safety. As an extreme example, suppose a program has one unsafe symbolic trajectory, but the probability of this symbolic trajectory is near-zero. This trajectory is likely to be missed by the sampling process, leading to an artificially low value of $C^\#(\theta)$.

**Example (Cont.)** Consider the trajectories $\tau_1^\#$ and $\tau_2^\#$ in our running example. We have $p(\tau_1^\#) = p(t_2|\sigma_2)$, as the other transitions in $\tau_1^\#$ have probability 1. We compute $p(t_2|\sigma_2) = \frac{\mathbf{Vol}([-2,1])}{\mathbf{Vol}([-2,2])} = 0.75$. Thus, $p(\tau_1^\#) = 0.75$ and similarly, $p(\tau_2^\#) = 0.25$.

**Gradient Estimation.** Our ultimate goal is to compute the gradient $\nabla_\theta C^\#(\theta)$ of the approximate safety loss. At first sight, the classic REINFORCE estimator (Williams, 1992) or one of its relatives (Schulman et al., 2015) seems suitable for this task, given that they can differentiate an integral over sampled "paths". However, a subtlety in our setting is that the losses for individual paths (symbolic trajectories) is a function of $\theta$. This calls for a generalization of traditional REINFORCE-like estimators.

More precisely, we adapt the derivation of REINFORCE to our setting as follows:

$$
\begin{aligned}
\nabla_\theta(C^\#(\theta)) &= \nabla_\theta \mathbf{E}_{\tau^\# \sim p_\theta(\tau^\#)} Unsafe_\theta(\tau^\#) \\
&= \nabla_\theta \int_{\tau^\# \sim p_\theta(\tau^\#)} p_\theta(\tau^\#) Unsafe_\theta(\tau^\#) d\tau^\# \\
&= \int_{\tau^\# \sim p_\theta(\tau^\#)} p_\theta(\tau^\#) \nabla_\theta Unsafe_\theta(\tau^\#) + Unsafe_\theta(\tau^\#) \nabla_\theta p_\theta(\tau^\#) d\tau^\# \\
&= \mathbf{E}_{\tau^\# \sim p_\theta(\tau^\#)} [\nabla_\theta Unsafe_\theta(\tau^\#)] + \mathbf{E}_{\tau^\# \sim p_\theta(\tau^\#)} [Unsafe_\theta(\tau^\#) \nabla_\theta(\log p_\theta(\tau^\#))]
\end{aligned}
$$

The above derivation can be further refined to incorporate the variance reduction techniques for REINFORCE that are commonly employed in generative modeling and reinforcement learning. The use of such techniques is orthogonal to our main contribution, and we ignore it in this paper.

## 5 EVALUATION

Our experimental evaluation seeks to answer the following research questions:

(**RQ1**): Is DSE effective in learning safe and performant parameters?

(**RQ2**): How does data size influence DSE and baselines?

(**RQ3**): How well does DSE scale as the program being learned gets more complex?

### 5.1 EXPERIMENTAL SETUP

**System Setup.** Our framework is built on top of PyTorch (Paszke et al., 2019). We use the Adam Optimizer (Kingma & Ba, 2014) for all the experiments with default parameters and a weight decay of 0.000001. We ran all the experiments using a single-thread implementation on a Linux system with Intel Xeon Gold 5218 2.30GHz CPUs and GeForce RTX 2080 Ti GPUs. (Please refer to Appendix A.7 for more training details.)

**Baselines.** We use two types of baselines: (i) *Ablation*, which ignores the safety constraint and learns only from data; (ii) DIFFAI+, an extended version of the original DIFFAI method (Mirman et al., 2018). DIFFAI does not directly fit our setting of neurosymbolic programs as it does not handle general conditional statements. DIFFAI+ extends DIFFAI by adding the meet and join operations following the abstract interpretation framework (Cousot & Cousot, 1977). Also, we allow DIFFAI+ to split the input set into several (100 if not specified) initial symbolic states to increase the precision of symbolic execution of each trajectory. (The original DIFFAI does not need to perform such splits because, being focused on adversarial robustness, it only propagates small regions around each input point through the model.) However, the parts of DIFFAI that propagate losses and their gradients through neural networks remain the same in DIFFAI+.

We use the box (interval) domain for both DSE and DIFFAI+ in all the experiments. DSE samples 50 symbolic trajectories for each starting component.

| Template | Model | Provably Safe Portion | |
| | | DIFFAI+ | DSE |
|---|---|---|---|
| Pattern1 | NNSmall | 0.0 | 1.0 |
| | NNMed | 0.0 | 1.0 |
| | NNBig | 0.0 | 1.0 |
| Pattern2 | NNSmall | 0.0 | 0.70 |
| | NNMed | 0.0 | 0.78 |
| | NNBig | 0.0 | 0.73 |
| Pattern3 | NNSmall | 1.0 | 1.0 |
| | NNMed | 1.0 | 1.0 |
| | NNBig | 1.0 | 1.0 |
| Pattern4 | NNSmall | 0.0 | 0.76 |
| | NNMed | 0.0 | 0.80 |
| | NNBig | 0.58 | 0.97 |
| Pattern5 | NNSmall | 0.0 | 1.0 |
| | NNMed | 0.0 | 1.0 |
| | NNBig | 0.0 | 1.0 |

Figure 2: Results of synthetic microbenchmarks of DSE and DIFFAI+.

**Benchmarks.** Our evaluation uses 5 synthetic microbenchmarks and 4 case studies. The microbenchmarks (Appendix A.5) are small programs consisting of basic neural network modules, arithmetic function assignment, and conditional branches. These programs are designed to highlight the relationship between DIFFAI+ and DSE. The case studies, described below, model real-world control and navigation tasks (see Appendix A.4 for more details):

- In *Thermostat*, we want to learn safe parameters for two neural networks that control the temperature of a room.

- *Racetrack* is a navigation benchmark (Barto et al., 1995; Christakis et al., 2021). Two vehicles are trained by path planners separately. The racetrack system is expected to learn two safe controllers so that vehicles do not crash into walls or into each other.

- In *Aircraft-Collision* (AC), the goal is to learn safe parameters for an airplane that performs maneuvers to avoid a collision with a second plane.

- *Cartpole* aims to train a cart imitating the demonstrations and keep the cart within the safe position range.

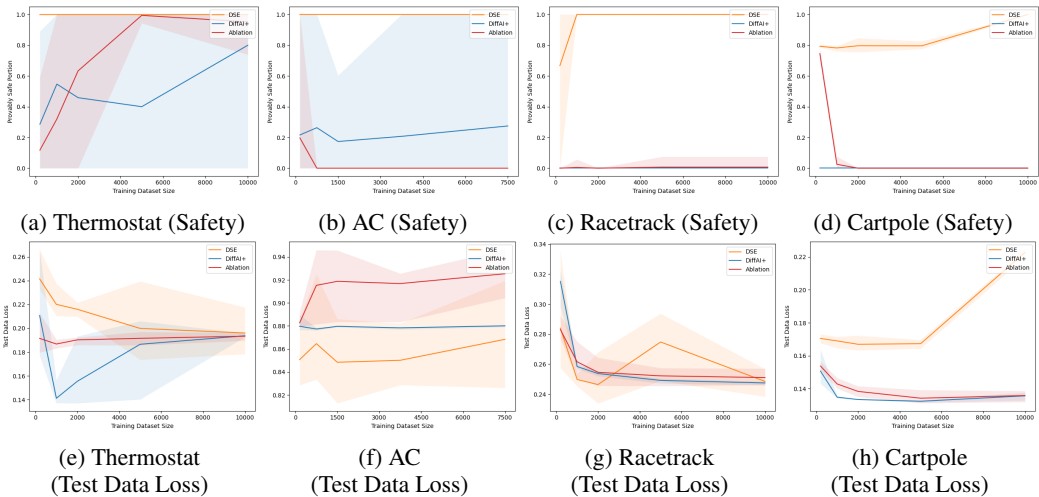

(a) Thermostat (Safety)    (b) AC (Safety)    (c) Racetrack (Safety)    (d) Cartpole (Safety)

(e) Thermostat
(Test Data Loss)

(f) AC
(Test Data Loss)

(g) Racetrack
(Test Data Loss)

(h) Cartpole
(Test Data Loss)

Figure 3: The provably safe portion and the test data loss of Ablation, DIFFAI+ and DSE when varying the size of the data to train.

For each benchmark, we first write a ground-truth program that does not include neural networks and provably satisfies a safety requirement. We identify certain modules of these programs as candidates for learning. Then we replace the modules with neural networks with trainable parameters.

For all benchmarks with the exception of Racetrack, we execute the ground-truth programs on uniformly sampled inputs to collect input-output datasets for each module that is replaced by a neural network. As for Racetrack, this benchmark involves two simulated vehicles with neural controllers; learned parameters for these controllers are safe if the vehicles do not crash into each other or a wall. To model real-world navigation, we generate the training data for the controllers from a ground-truth trajectory constructed using a path planner. This path planner only tries to avoid the walls in the map and does not have a representation of distances from other vehicles. Thus, in this problem, even a very large data set does not contain all the information needed for learning safe parameters.

**Evaluation of Safety and Data Loss.** Once training is over, we evaluate the learned program's *test data loss* by running it on 10000 initial states that were not seen during training. We evaluate the safety loss using an abstract interpreter (Cousot & Cousot, 1977) that splits the initial condition into a certain number of boxes ($20^4$ for Cartpole, 10000 for the other benchmarks), then constructs symbolic trajectories from these boxes. This analysis is sound (unlike the approximate loss employed during DSE training), meaning that the program is provably worst-case safe if the safety loss evaluates to 0. Our safety metric is the *provably safe portion*, which is the fraction of the program's symbolic trajectories that are worst-case safe.

## 5.2 RESULTS

**RQ1: Overall Quality of Learned Parameters.** Figure 2 exhibits the provably safe portion of DIFFAI+ and DSE on our microbenchmarks. Here, the first two patterns only use neural modules to evaluate the guards of a conditional branch, and the last three benchmarks use the neural modules in the guards as well as updates (assignments) that directly affect the program's safety. We see that DIFFAI+ cannot learn safe highly non-differentiable modules (pattern1, pattern2) while DSE can. For the cases where optimization across branches is not required to give safe programs, DIFFAI+ can handle them (pattern3). For the patterns with neural assignments, DIFFAI+ finds it hard to learn to jump from one conditional branch to another to increase safety (pattern4, pattern5). We refer the readers to Appendix A.9 for more detailed pattern analysis.

We show the training-time and test-time safety and data losses for our more complex benchmarks in Figure 3. From Figure 3a, 3b, 3c, 3d, we exhibit that DSE can learn programs with 0.8 provably safe portion even with 200 data points and the results keep when the number of training data points is increased. Meanwhile, both DIFFAI+ and Ablation fail to provide safe programs for AC, Racetrack and Cartpole. For Thermostat, Ablation and DIFFAI+ can achieve a provably safe portion of 0.8 only when using 5000 and 10000 data points to train.

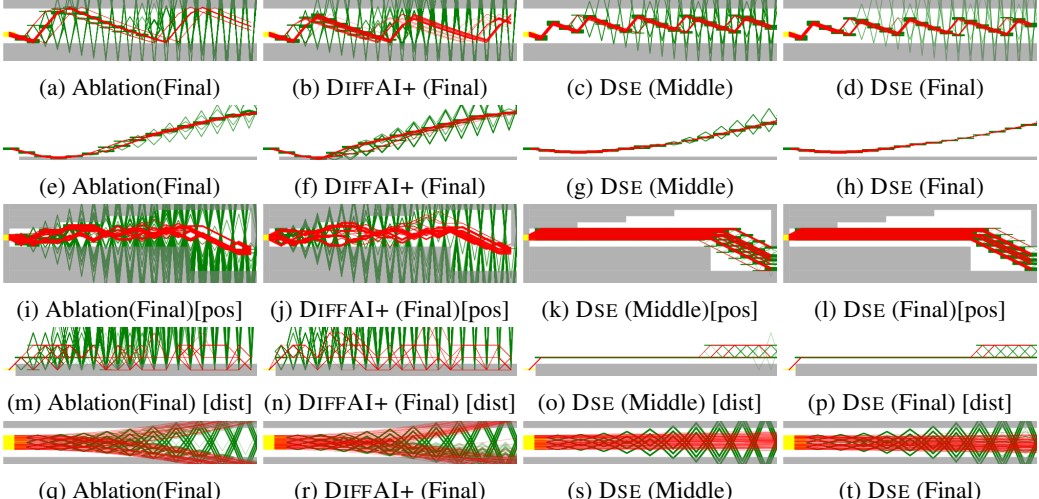

| (a) ABLATION(Final) | (b) DIFFAI+ (Final) | (c) DSE (Middle) | (d) DSE (Final) |
| (e) ABLATION(Final) | (f) DIFFAI+ (Final) | (g) DSE (Middle) | (h) DSE (Final) |
| (i) ABLATION(Final)[pos] | (j) DIFFAI+ (Final)[pos] | (k) DSE (Middle)[pos] | (l) DSE (Final)[pos] |
| (m) ABLATION(Final) [dist] | (n) DIFFAI+ (Final) [dist] | (o) DSE (Middle) [dist] | (p) DSE (Final) [dist] |
| (q) ABLATION(Final) | (r) DIFFAI+ (Final) | (s) DSE (Middle) | (t) DSE (Final) |

Figure 4: Trajectories during training. Each row exhibits the concrete trajectories and symbolic trajectories of one case from different methods. From top to bottom, the cases are Thermostat (Figure. 4a, 4b, 4c, 4d), AC (Figure. 4e, 4f, 4g, 4h), Racetrack with position property (Figure. 4i, 4j, 4k, 4p) and distance property (Figure. 4m, 4n, 4o, 4p), and Cartpole (Figure. 4q, 4r, 4s, 4t). Each figure shows the trajectories (concrete: red, symbolic: green rhombus) of programs learnt by the method from different training stages, which is denoted by "Middle" and "Final". We separate the input state set into 100 components (81 components for Cartpole) evenly to plot the symbolic trajectories clearly. During evaluation, we separate the input set into 10000 components ($20^4$ components for Cartpole) evenly to get more accurate symbolic trajectories measurement.

Our test data loss is sometimes larger yet comparable with the Ablation and DIFFAI+ from Figure 3e, 3g. Figure 3h exhibits the tension between achieving a good loss and safety, where the test data loss of DSE increases as the provably safe portion becomes larger. For AC specifically, the safety constraint can help the learner overcome some local optimal yet unsafe areas to get a safer result with more accurate behaviors.

Overall, we find that DSE outperforms DIFFAI+ when neural modules are used in the conditional statement and the interaction between neural modules is important for the safety property. This is the case in our larger benchmarks. For example, in Thermostat and AC, the neural modules' outputs decide the actions that the neural module to take in the next steps. In Racetrack, the distance between two neural modules regulates each step of the vehicles' controllers. In Cartpole, the neural modules gives the force on carts as the output.

We display representative symbolic trajectories from the programs learned by Ablation, DIFFAI+ and DSE in Figure 4. The larger portion of the symbolic trajectories is provably safe from DSE than the ones from baselines, which is indicated by the less overlapping between the green trajectories and the gray area. Because the symbolic trajectories are overapproximate, we also depict 100 randomly sampled representative concrete trajectories for each approach. We note that more concrete trajectories of Thermostat, AC, the distance property of Racetrack and Cartpole from DSE fall into the safe area compared with DIFFAI+ and Ablation.

**RQ2: Impact of Data Size.** Figure 3 compares the different methods' performance on the test metrics as the number of training data points is changed. The Ablation and DIFFAI+ can reach 0.95 provably safe portion with 10000 data points for Thermostat. However, the variance of results is large, as the minimum provably safe portion across training can reach 0.74 and 0.0 for Ablation and DIFFAI+. In the Thermostat benchmark, DIFFAI+'s performance mainly comes from the guidance from the data loss rather than the safety loss.

For the other three cases, a larger training data set cannot help the Ablation and DIFFAI+ to give much safer programs. For AC, as is in Figure 4e, the program learnt from Ablation is very close to the safe area but the Ablation is still not accurate enough to give a safe program in the third step. In Racetrack, increasing the data size to train the vehicles' controllers can only satisfy the requirement of "not crashing into walls". The networks cannot learn to satisfy the property of not crashing into

other vehicles as this information is not available in the training data. In Cartpole, the baselines mainly follow the guidance of the data loss and can not give safer programs.

**RQ3: Scalability.** As our solution for the safety loss does not rely on the number of data points, the solution is scalable in terms of both data size and neural network size. In Figure 2, we use the three types of neural networks: (i) NNSmall (with about 33000 parameters); (ii) NNMed (with over 0.5 million parameters); (iii) NNBig (with over 2 million parameters), the training time per epoch is 0.09 sec, 0.10sec, 0.11sec for NNSmall, NNMed, NNBig separately. For the synthetic benchmarks, the program can converge within about 200 epochs which takes 18 sec, 20sec, and 22 sec.

Our Thermostat, AC and Racetrack benchmarks consist of neural networks with 4933, 8836 and 4547 parameters respectively, and loops with 20, 15 and 20 iterations respectively. There are 2, 1 and 2 neural modules (respectively) in each benchmark. The training times for DIFFAI+ and DSE are comparable on these benchmarks. Specifically, for Thermostat, AC, and Racetrack, the total training time of DSE is less than 1 hour, less than 90 minutes, and around 13 hours. The corresponding numbers are over 90 minutes, more than 2 hours, and more than 17 hours for DIFFAI+. A more detailed scalability analysis is available in Appendix A.11.

## 6 RELATED WORK

**Verification of Neural and Neurosymbolic Models.** There are many recent papers on the verification of worst-case properties of neural networks (Anderson et al., 2019; Gehr et al., 2018; Katz et al., 2017; Elboher et al., 2020; Wang et al., 2018). There are also several recent papers (Ivanov et al., 2019; Tran et al., 2020; Sun et al., 2019; Christakis et al., 2021) on the verification of compositions of neural networks and symbolic systems (for example, plant models). To our knowledge, the present effort is the first to integrate a method of this sort — propagation of worst-case intervals through neurosymbolic program — with the gradient-based training of neurosymbolic programs.

**Verified Deep Learning.** There is a growing literature on methods that incorporate worst-case objectives (such as safety and robustness) into neural network training (Zhang et al., 2019; Mirman et al., 2018; Madry et al., 2017; Cohen et al., 2019; Singh et al., 2018). Most work on this topic focuses on the training of single neural networks. There are a few domain-specific efforts that consider the environment of the neural networks being trained. For example, (Shi et al., 2019) uses spectral normalization to constrain the neural network module of a neurosymbolic controller and ensure that it respects certain stability properties. Unlike DSE, these methods treat neural network modules in an isolated way and do not consider the interactions between these modules and surrounding code (Stone et al., 2016; Qin & Badgwell, 2000; Nassi et al., 2020; Katz et al., 2021).

**Verified Parameter Synthesis for Symbolic Code.** There is a large body of work on parameter synthesis for traditional symbolic code (Alur et al., 2013; Chaudhuri et al., 2014) with respect to worst-case correctness constraints. Especially relevant in this literature is Chaudhuri et al. (2014), which introduced a method to smoothly approximate abstract interpreters of programs that is closely related to DSE and DIFFAI. However, these methods do not use contemporary gradient-based learning, and as a result, scaling them to programs with neural modules is impractical.

## 7 CONCLUSION

We presented DSE, the first approach to worst-case-safe parameter learning for programs that embed neural networks inside potentially discontinuous symbolic code. The method is based on a new mechanism for differentiating through a symbolic executor. We demonstrate that DSE outperforms DIFFAI, a state-of-the-art approach to worst-case-safe deep learning, in a range of tasks.

Our current implementation of DSE uses an interval representation of symbolic states. Future work should explore more precise representations such as zonotopes. Incorporating modern variance reduction techniques in our gradient estimator is another natural next step. Finally, one challenge in DSE is that learning here can get harder as the symbolic state representation gets more precise. In particular, if we increase the number of initial symbolic states beyond a point, each state would only lead to a unique symbolic trajectory, and there would be no gradient signal to adjust the relative weights of the different symbolic trajectories. Future work should seek to identify good, possibly adaptive, tradeoffs between the precision of symbolic states and the ease of learning.

**Acknowledgments:** We thank our anonymous reviewers for their insightful comments and Radoslav Ivanov (Rensselaer Polytechnic Institute) for his help with benchmark design. Work on this paper was supported by the United States Air Force and DARPA under Contract No. FA8750-20-C-0002, by ONR under Award No. N00014-20-1-2115, and by NSF under grant #1901376.

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

---

**Algorithm 1:** Learning Safe, Optimal Parameter Mixtures (Agarwal et al., 2018; Le et al., 2019)

---

**for** $t = 1, ..., N$ **do**
    $\theta_t \leftarrow \mathbf{Best}_\theta(\lambda_t)$
    $\hat{\theta}_t \leftarrow \mathbf{Uniform}(\theta_1, \ldots, \theta_t), \ \hat{\lambda}_t \leftarrow \frac{1}{t} \sum \lambda_t$
    $L_{\max} \leftarrow L(\hat{\theta}, \mathbf{Best}_\lambda(\hat{\theta}))$
    $L_{\min} \leftarrow L(\mathbf{Best}_\theta(\hat{\lambda}_t), \hat{\lambda}_t)$
    **if** $L_{\max} - L_{\min} < \nu$ **then return** $(\hat{\theta}_t, \hat{\lambda}_t)$;
    $\lambda_{t+1} \leftarrow \boldsymbol{\lambda}\text{-}\mathbf{Update}(\theta_1, \ldots, \theta_t)$

---

# A  APPENDIX

## A.1  LEARNING FRAMEWORK

We use an equivalence between constrained and regularized learning that (Agarwal et al., 2018; Le et al., 2019), among others, have recently developed in other learning settings, we reduce our problem to a series of unconstrained optimization tasks.

We convexify the program set $\{F_\theta : \theta \in \mathbb{R}^k\}$ by considering stochastic mixtures (Le et al., 2019) and represent the convexified set as a probabilistic function $F_{\hat{\theta}}$. Following DSE, we rewrite Equation 2 in terms of these mixtures:

$$\hat{\theta}^* \quad = \quad \underset{\hat{\theta}}{\arg\min} \, Q(\hat{\theta}) \qquad \text{s.t. } C^{\#}(\hat{\theta}) \leq 0 \tag{6}$$

We convert Equation 6 to a Lagrangian function

$$L(\hat{\theta}, \lambda) = Q(\hat{\theta}) + \lambda C^{\#}(\hat{\theta}) \tag{7}$$

$$\tag{8}$$

, where $\lambda \in \mathbb{R}^+$ is a *Lagrange multiplier*. Following the equilibrium computation technique by Freund & Schapire (1996), Equation 6 can be rewritten as

$$\max_{\lambda \in \mathbb{R}^+} \min_{\hat{\theta}} L(\hat{\theta}, \lambda). \tag{9}$$

Solutions to this problem can be interpreted as equilibria of a game between a $\lambda$-player and a $\hat{\theta}$-player in which the $\hat{\theta}$-player minimizes $L(\hat{\theta}, \lambda)$ given the current $\lambda$, and the $\lambda$-player maximizes $L(\hat{\theta}, \lambda)$ given the current $\hat{\theta}$. Our overall algorithm is shown in Algorithm 1. In this pseudocode, $\nu$ is a predefined positive real. $\mathbf{Uniform}(\theta_1, \ldots, \theta_t)$ is the mixture that selects a $\theta_i$ out of $\{\theta_1, \ldots, \theta_t\}$ uniformly at random.

$\mathbf{Best}_\theta(\lambda)$ refers to the $\hat{\theta}$ player's *best response* for a given value of $\lambda$ (it can be shown that this best response is a single parameter value, rather than a mixture of parameters). Computing this best response amounts to solving the unconstrained optimization problem $\min_\theta L(\theta, \lambda)$, i.e.,

$$\mathbf{Best}_\theta(\lambda) = \min_\theta Q(\theta) + \lambda C^{\#}(\theta). \tag{10}$$

$\mathbf{Best}_\lambda(\hat{\theta})$ is the $\lambda$-player's best response to a particular parameter mixture $\hat{\theta}$. We define this function as

$$\mathbf{Best}_\lambda(\hat{\theta}) = \begin{cases} 0 & \text{if } C^{\#}(\hat{\theta}) \leq 0 \\ S & \text{otherwise} \end{cases} \tag{11}$$

where $S$ is the upper bound on $\lambda$. Intuitively, when $C^{\#}(\hat{\theta})$ is non-positive, the current parameter mixture is safe. As $L$ is a linear function of $\lambda$, the minimum value of $L$ is achieved in this case when $\lambda$ is zero. For other cases, the minimum $L$ is reached by setting $\lambda$ to the maximum value $S$.

Finally, $\boldsymbol{\lambda}\text{-}\mathbf{Update}(\theta_t)$ computes, in constant time, a new value of $\lambda$ based on the most recent best response by $\hat{\theta}$ player. We do not describe this function in detail; please see (Agarwal et al., 2018) for more details.

Following prior work, we can show that Algorithm 1 converges to a value $(\hat{\theta}, \hat{\lambda})$ that is within additive distance $\nu$ from a saddle point for Equation 9. To solve our original problem (Equation 2),

we take the returned $\hat{\theta}$ and then return the real-valued parameter $\theta_i$ to which this mixture assigns the highest probability. It is easy to see that this value is an approximate solution to Equation 2.

## A.2 Abstract Update for Neural Network Modules

We consider the box domain in the implementation. For a program with $m$ variables, each component in the domain represents a $m$-dimensional box. Each component of the domain is a pair $b = \langle b_c, b_e \rangle$, where $b_c \in \mathbb{R}^m$ is the center of the box and $b_e \in \mathbb{R}^m_{\geq 0}$ represents the non-negative deviations. The interval concretization of the $i$-th dimension variable of $b$ is given by

$$[(b_c)_i - (b_e)_i, (b_c)_i + (b_e)_i].$$

Now we give the abstract update for the box domain following Mirman et al. (2018).

**Add.** For a concrete function $f$ that replaces the $i$-th element in the input vector $x \in \mathbb{R}^m$ by the sum of the $j$-th and $k$-th element:

$$f(x) = (x_1, \ldots, x_{i-1}, x_j + x_k, x_{i+1}, \ldots x_m)^T.$$

The abstraction function of $f$ is given by:

$$f^\#(b) = \langle M \cdot b_c, M \cdot b_e \rangle,$$

where $M \in \mathbb{R}^{m \times m}$ can replace the $i$-th element of $x$ by the sum of the $j$-th and $k$-th element by $M \cdot b_c$.

**Multiplication.** For a concrete function $f$ that multiplies the $i$-th element in the input vector $x \in \mathbb{R}^m$ by a constant $w$:

$$f(x) = (x_1, \ldots, x_{i-1}, w \cdot x_i, x_{i+1}, \ldots, x_m)^T.$$

The abstraction function of $f$ is given by:

$$f^\#(b) = \langle M_w \cdot b_c, M_{|w|} \cdot b_e \rangle,$$

where $M_w \cdot b_c$ multiplies the $i$-th element of $b_c$ by $w$ and $M_{|w|} \cdot b_e$ multiplies the $i$-th element of $b_e$ with $|w|$.

**Matrix Multiplication.** For a concrete function $f$ that multiplies the input $x \in \mathbb{R}^m$ by a fixed matrix $M \in \mathbb{R}^{m' \times m}$:

$$f(x) = M \cdot x.$$

The abstraction function of $f$ is given by:

$$f^\#(b) = \langle M \cdot b_c, |M| \cdot b_e \rangle,$$

where $M$ is an element-wise absolute value operation. Convolutions follow the same approach, as they are also linear operations.

**ReLU.** For a concrete element-wise ReLU operation over $x \in \mathbb{R}^m$:

$$\text{ReLU}(x) = (\max(x_1, 0), \ldots, \max(x_m, 0))^T,$$

the abstraction function of ReLU is given by:

$$\text{ReLU}^\#(b) = \langle \frac{\text{ReLU}(b_c + b_e) + \text{ReLU}(b_c - b_e)}{2}, \frac{\text{ReLU}(b_c + b_e) - \text{ReLU}(b_c - b_e)}{2} \rangle.$$

where $b_c + b_e$ and $b_c - b_e$ denotes the element-wise sum and element-wise subtraction between $b_c$ and $b_e$.

**Sigmoid.** As Sigmoid and ReLU are both monotonic functions, the abstraction functions follow the same approach. For a concrete element-wise Sigmoid operation over $x \in \mathbb{R}^m$:

$$\text{Sigmoid}(x) = (\frac{1}{1 + \exp(-x_1)}, \ldots, \frac{1}{1 + \exp(-x_m)})^T,$$

the abstraction function of Sigmoid is given by:

$$\text{Sigmoid}^\#(b) = \langle \frac{\text{Sigmoid}(b_c + b_e) + \text{Sigmoid}(b_c - b_e)}{2}, \frac{\text{Sigmoid}(b_c + b_e) - \text{Sigmoid}(b_c - b_e)}{2} \rangle.$$

where $b_c + b_e$ and $b_c - b_e$ denotes the element-wise sum and element-wise subtraction between $b_c$ and $b_e$. All the above abstract updates can be easily differentiable and parallelized on the GPU.

## A.3 INSTANTIATION OF THE *Unsafe* FUNCTION

In general, the $Unsafe(s)$ function over individual states can be any differentiable distance function between a point and a set which satisfies the property that $Unsafe(s) = 0$ if $s$ is in the safe set, $\mathcal{A}$, and $Unsafe(s) > 0$ if $s$ is not in $\mathcal{A}$. We give the following instantiation as the unsafeness score over individual states:

$$Unsafe(s) = \begin{cases} \min_{x \in \mathcal{A}} \text{DIST}(s, x) & \text{if } s \notin \mathcal{A} \\ 0 & \text{if } s \in \mathcal{A} \end{cases}$$

where DIST denotes the euclidean distance between two points.

Similarly, the $Unsafe(\sigma)$ function over symbolic states can be any differentiable distance function between two sets which satisfies the property that $Unsafe(\sigma) = 0$ if $V$ is in $\mathcal{A}$, and $Unsafe(\sigma) > 0$ if $V$ is not in $\mathcal{A}$. We give the following instantiation as a differentiable approximation of the worst-case loss $\max_{s \text{ satisfies } V} Unsafe(s)$ in our implementation:

$$Unsafe(\sigma) = \begin{cases} \min_{s \text{ satisfies } V} Unsafe(s) + 1 & \text{if } V \wedge \mathcal{A} = \emptyset \\ 1 - \frac{\mathbf{Vol}(V \wedge \mathcal{A})}{\mathbf{Vol}(V)} & \text{if } V \wedge \mathcal{A} \neq \emptyset \end{cases}$$

A.4   BENCHMARKS FOR CASE STUDIES

The detailed programs describing Aircraft Collision, Thermostat and Racetrack are in Figure 5, 6 and 7. The $\pi_\theta$ of Aircraft Collision and $\pi_\theta^{\text{cool}}$, $\pi_\theta^{\text{heat}}$ in Thermostat are a 3 layer feed forward net with 64 nodes in each layer and a ReLU after each layer except the last one. A sigmoid layer serves as the last layer. Both the $\pi_\theta^{\text{agent1}}$ and $\pi_\theta^{\text{agent2}}$ in Racetrack are a 3 layer feed forward net with 64 nodes in each and a ReLU after each layer. The Cartpole benchmark is a linear approximation following Bastani et al. (2018) with a 3 layer feed forward net with 64 nodes in each layer as the controller.

```
1   aircraftcollision(x):
2       x1, y1 := x, -15.0
3       x2, y2 := 0.0, 0.0
4       N := 15
5       i = 0
6       while i < N:
7           p0, p1, p2, p3, step := πθ(x1, y1, x2, y2, step, stage)
8           stage := argmax(p0, p1, p2, p3)
9           if stage == CRUISE:
10              x1, y1 := MOVE_CRUISE(x1, y1)
11          else if stage == LEFT:
12              x1, y1 := MOVE_LEFT(x1, y1)
13          else if stage == STRAIGHT:
14              x1, y1 := MOVE_STRAIGHT(x1, y1)
15          else if stage == RIGHT:
16              x1, y1 := MOVE_RIGHT(x1, y1)
17
18          x2, y2 := MOVE_2(x2, y2)
19          i := i + 1
20          assert (!IS_CRASH(x1, y1, x2, y2))
21      return
```

Figure 5: Aircraft Collision

```
1   thermostat(x):
2       N = 20
3       isOn = 0.0
4       i = 0
5       while i < N:
6           if isOn ≤ 0.5:
7               isOn := πθ^cool(x)
8               x := COOLING(x)
9           else:
10              isOn, heat := πθ^heat(x)
11              x := WARMING(x, heat)
12          i := i + 1
13          assert(!EXTREME_TEMPERATURE(x))
14
15      return
```

Figure 6: Thermostat

```
1   racetrack(x):
2       x1, y1, x2, y2 := x, 0.0, x, 0.0
3       N := 20
4       while i < N:
5           p10, p11, p12 := π_θ^agent1(x1, y1)
6           p20, p21, p22 := π_θ^agent2(x2, y2)
7           action1 := argmax(p10, p11, p12)
8           action2 := argmax(p20, p21, p22)
9           x1, y1 := MOVE(x1, y1, action1)
10          x2, y2 := MOVE(x2, y2, action2)
11          i := i + 1
12          assert(!CRASH_WALL(x1, y1) && !CRASH_WALL(x2, y2)
13              && !CRASH(x1, y1, x2, y2))
14
15      return
```

Figure 7: Racetrack

## A.5 SYNTHETIC MICROBENCHMARKS

Figure 8 exhibits the 5 synthetic microbenchmarks in our evaluation. Specifically, the neural network modules are used as conditions in pattern 1 and pattern 2. The neural network modules serve as both conditions and the assignment variable in pattern 3, pattern 4 and pattern 5. To highlight the impact from the safety loss, we only train with safety loss for synthetic microbenchmarks.

```
1  pattern1(x):
2      # x ∈ [-5, 5]
3      y := π_θ(x)
4      if y ≤ 1.0:
5          z := 10
6      else:
7          z: = 1
8      assert(z ≤ 1)
9      return z
```

(a) Pattern1

```
1  pattern2(x):
2      # x ∈ [-5, 5]
3      y := π_θ(x)
4      if y ≤ 1.0:
5          z := x + 10
6      else:
7          z: = x - 5
8      assert(z ≤ 0)
9      return z
```

(b) Pattern2

```
1  pattern3(x):
2      # x ∈ [-5, 5]
3      y := π_θ(x)
4      if y ≤ 1.0:
5          z := 10 - y
6      else:
7          z: = 1
8      assert(z ≤ 1)
9      return z
```

(c) Pattern3

```
1  pattern4(x):
2      # x ∈ [-5, 5]
3      y := π_θ(x)
4      if y ≤ -1.0:
5          z := 1
6      else:
7          z: = 2 + y * y
8      assert(z ≤ 1)
9      return z
```

(d) Pattern4

```
1  pattern5(x):
2      # x ∈ [-1, 1]
3      y := π_θ(x)
4      if y ≤ 1.0:
5          z := y
6      else:
7          z: = -10
8      assert(z ≤ 0 && z ≥ -5)
9      return z
```

(e) Pattern5

Figure 8: Programs for Patterns

A.6   DATA GENERATION

In this section, we provide more details about our data generation process. We give our ground-truth programs and the description of each benchmark:

- Thermostat: Figure 6 exhibits the program with initial temperature $x \in [60.0, 64.0]$ in training, where a 20-length loop is involved. The COOLING and WARMING are two differentiable functions updating the temperature, $x$. The $\pi_\theta^{cool}$ and $\pi_\theta^{heat}$ are two linear feed-forward neural networks. We set the safe temperature to the area $[55.0, 83.0]$. EXTREME_TEMPERATURE measures that whether $x$ is not within the safe temperature scope. *The ground-truth program replaces $\pi_\theta^{cool}$ and $\pi_\theta^{heat}$ with two different functional mechanisms (including branches and assignment), which decides the value of isOn and heat.* To mimic the real-world signal sensing process of thermostat, we carefully add noise to the output of these functional mechanisms and restricts the noise does not influence the safety of the ground-truth program.

- AC: Figure 5 illustrates the program in training with initial input x-axis of aircraft1 $x \in [12.0, 16.0]$, where a 15-length loop is involved. The MOVE_* functions update the position of aircraft in a differentiable way. The $\pi_\theta$ is one linear feed-forward neural networks. IS_CRASH measures the distances between two aircraft and the safe distance area is set to be larger than 40.0. *In the ground-truth program, the $\pi_\theta$ is replaced by one functional mechanism mimicing the real-world aircraft planner.* If the stage is CRUISE, the ground-truth planner detects the distance between aircraft and decides the next step's stage by assignment values 1 or 0 to p0, p1, p2, p3. The step is set to 0. When the stage is in LEFT or RIGHT, stage keeps and the step increases if the number of steps in this stage is within a threshold, and the stage changes to STRAIGHT or CRUISE and the step is set to 0 otherwise. Specifically, we convert the argmax to a nested If-Then-Else(ITE) block to select the index of the $p$ with the maximum value.

- Racetrack: Figure 7 illustrates the program in training with the input x-axis of each agent $x \in [5.0, 6.0]$ (The two agents start from the same location.) This program has a 20-length loop with a dynamic length of 223 lines (The unfold ITE of argmax consists of 3 lines code.) In training, the $\pi_\theta^{agent1}$ and $\pi_\theta^{agent2}$ are two linear feed-forward neural network. The MOVE is differentiable functions that update agents' positions. $CRASH\_WALL$ measures whether the agents' position collides with the wall. $CRASH$ measures whether the two agents crash into each other. *In ground-truth program, $\pi_\theta^{agent1}$ and $\pi_\theta^{agent2}$ are replaced by functional path planners guiding the agent's direction. The path planner only tries to avoid wall-collision in the map. That is to say, in the trajectories data generated, an arbitrary pair of trajectories of agent1 and agent2 is not guaranteed to be no-collision with each other.* To model real-world navigation, we also add noise in the next step selection for one position to generate independent trajectories of each agent. In the ground-truth program, the noise is added by uniformly selecting from the safe next step area.

- Cartpole: In the Cartpole experiment, we use the trajectories data generated from the expert model from Imitation Package. Starting from $S_0 = [-0.05, 0.05]^4$, we train a cart of which the position is within a range $x \in [-0.1, 0.1]$. The state space is 4 and the action space is 2. We use a 3 layer fully connected neural network (each layer followed by a ReLU and the last layer followed by a Sigmoid) to train. We follow the two points in Bastani et al. (2018) below to setup the experiment:
    - We approximate the system using a finite time horizon $T_{max} = 10$;
    - we use a linear approximation f(s, a) $\simeq$ As + Ba.

When generating the trajectory dataset, we uniformly sample the input from the input space and run the ground-truth program to get the trajectory. Specifically, the trajectories in the dataset is represented by a sequence of neural network input-output pairs and the index of the neural networks. Take Thermostat as an example, we get a sequence of $\langle ([x_0], [isOn_0], ``cool"), \dots, ([x_k], [isOn_k, heat_k], ``heat"), \dots \rangle$ for one starting point. The $k$ represents the data collected in the $k$-th step of the program.

## A.7 TRAINING PROCEDURE

Our framework is built on top of PyTorch (Paszke et al., 2019). We use the Adam Optimizer (Kingma & Ba, 2014) for all the experiments with a learning rate of $0.001$ and a weight decay of $0.000001$. We ran all the experiments using a single-thread implementation on a Linux system with Intel Xeon Gold 5218 2.30GHz CPUs and GeForce RTX 2080 Ti GPUs.

We represent the data loss ($Q$) following imitation learning techniques. The trajectory dataset is converted to several input-output example pair sets. Each input-output pair set represents the input-output pairs for each neural network. $Q$ is calculated by the average of the Mean Squared Error of the each neural network over the corresponding input-output pair set.

The $C^{\#}$ of DSE follows the algorithm given in the Section 4 with one symbolic state covering the input space. In DIFFAI+, we use sound join operation, which is used in classic abstract interpretation Cousot & Cousot (1977), whenever encountering branches. Sound join means that the symbolic state after the branch block is the join of the symbolic states computed from each branch. The concretization of the symbolic state after sound join covers the union of the concretization of the symbolic states from each branch. Intuitively, starting from one input symbolic state, DIFFAI+ gets one symbolic trajectory which covers all the potential trajectories starting from the concrete input satisfying the input symbolic state. For the safety loss of DIFFAI+, we first split the input space into 100 subregions evenly to give more accuracy for DIFFAI+. After that, we can get 100 symbolic trajectories. We compute the average of the safety loss of all the 100 symbolic trajectories as the training safety loss $C^{\#}$ of DIFFAI+.

We set the convergence criterion as a combination of epoch number and loss. In each training iteration, the final loss is represented as $Q(\theta) + C^{\#}(\theta)$. We set the maximum epoch for Thermostat, AC and Racetrack as 1500, 1200 and 6000. We set the early stop if the final loss has less than a 1% decrease over 200 epochs.

## A.8 TRAINING PERFORMANCE

Figure 9 and Figure 10 illustrates the training performance of safety loss and data loss when we vary data points (2%, 10%, 20%, 50%, 100% of the training dataset) used by DIFFAI + and DSE. We can see that the safety loss of DSE can converge while the variance of the safety loss of DIFFAI + is quite large for these three cases and can not converge to a small safety loss.

While we found that starting from random initialization can give us safe programs in Thermostat, we achieved much quicker convergence when initializing Thermostat programs trained purely with data and then continue to train with safety loss. We show the results and training performance for Thermostat with data-loss initialized program. The other two benchmarks do not benefit a lot from the data loss initialized program. We keep the training of them with random initialization.

### A.8.1 SAFETY LOSS

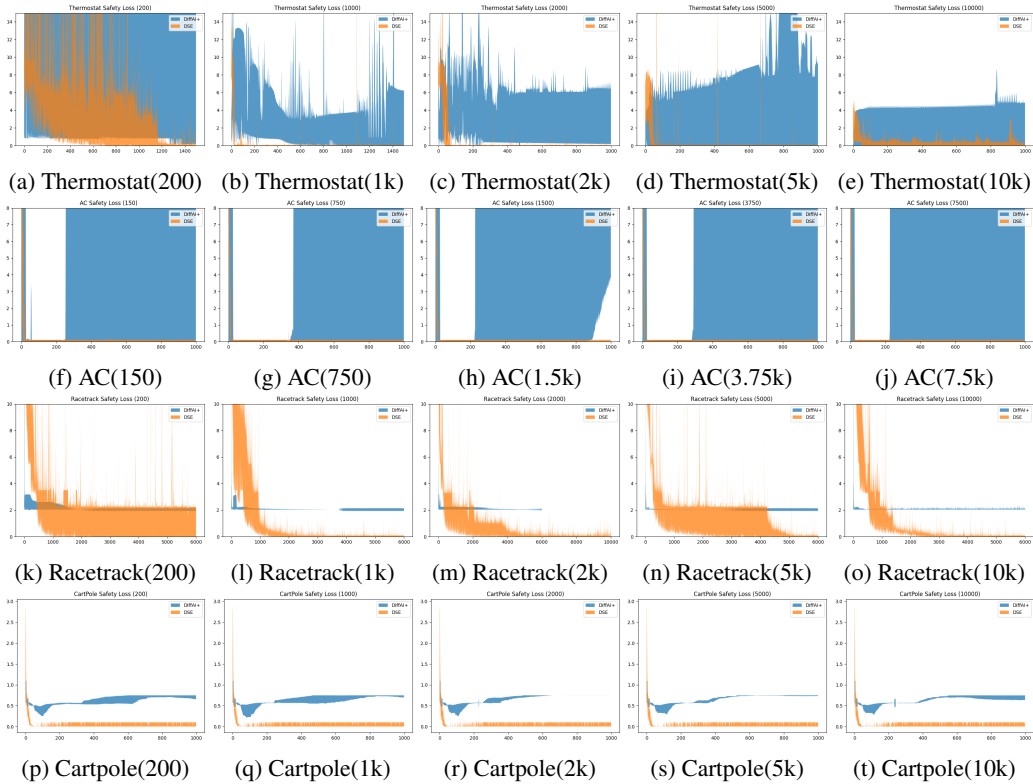

(a) Thermostat(200)  (b) Thermostat(1k)  (c) Thermostat(2k)  (d) Thermostat(5k)  (e) Thermostat(10k)

(f) AC(150)  (g) AC(750)  (h) AC(1.5k)  (i) AC(3.75k)  (j) AC(7.5k)

(k) Racetrack(200)  (l) Racetrack(1k)  (m) Racetrack(2k)  (n) Racetrack(5k)  (o) Racetrack(10k)

(p) Cartpole(200)  (q) Cartpole(1k)  (r) Cartpole(2k)  (s) Cartpole(5k)  (t) Cartpole(10k)

Figure 9: Training performance of data loss on different benchmarks varying data points. The y-axis represents the safety loss ($C^{\#}(\theta)$) and the x-axis gives the number of training epochs. Overall, DSE converges within 1500, 1000 , 6000, and 2000 epochs for Thermostat, AC, Racetrack, Cartpole, while DIFFAI+ easily gets stuck or fluctuates on these benchmarks.

## A.8.2 DATA LOSS

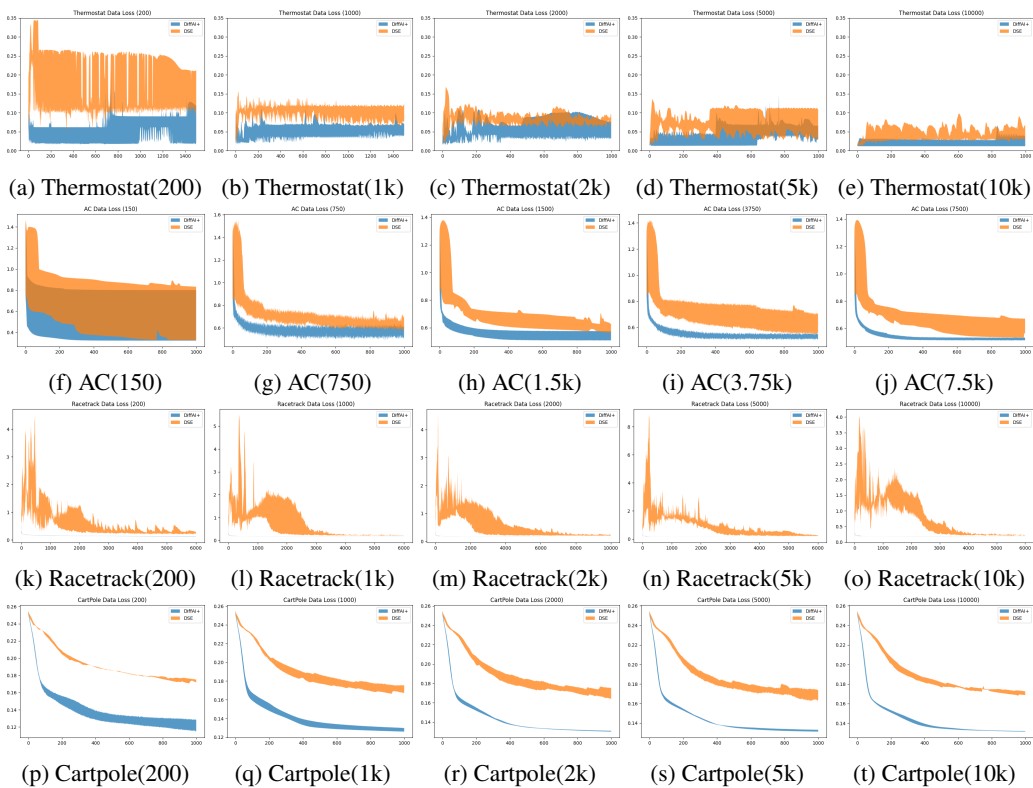

Figure 10: Training performance of data loss on different benchmarks varying data points. The y-axis represents the data loss ($Q(\theta)$) and the x-axis gives the number of training epochs. Since Thermostat is initialized with data trained programs to give quicker convergence, the data loss does not change a lot during training. For other benchmarks, data loss of DSE and DIFFAI+ are both converging. Overall speaking, DSE sacrifices some data loss to give safer programs.

## A.8.3 DETAILS OF TRAINING AND TEST DATA

We also attach the detailed training data loss ($Q$), training safety loss ($C^{\#}$), test data loss and provably safe portion for Thermostat(Figure 11), AC(Figure 12) and Racetrack(Figure 13). Specifically, training safety loss measures the expectation of the quantitative safety of the symbolic trajectories and provably safe portion measures the portion of the symbolic trajectories that is provably safe. Intuitively, in training, if the number of unsafe symbolic trajectories is larger and the unsafe symbolic trajectories are further away from safe areas, $C^{\#}$ is larger. In test, if the number of provably safe symbolic trajectories is larger, the provably safe portion is larger.

| Data Size | Approach | $Q$ | $C^{\#}$ | Test Data Loss | Provably Safe Portion |
|---|---|---|---|---|---|
| 200 | DSE | 0.13 | 0.02 | 0.24 | 0.99 |
|  | DIFFAI+ | 0.07 | 25.37 | 0.21 | 0.28 |
| 1000 | DSE | 0.09 | 0.0 | 0.22 | 0.99 |
|  | DIFFAI+ | 0.05 | 1.96 | 0.14 | 0.55 |
| 2000 | DSE | 0.08 | 0.0 | 0.21 | 0.99 |
|  | DIFFAI+ | 0.05 | 2.70 | 0.15 | 0.46 |
| 5000 | DSE | 0.07 | 0.0 | 0.19 | 0.99 |
|  | DIFFAI+ | 0.04 | 4.58 | 0.18 | 0.40 |
| 10000 | DSE | 0.04 | 0.0 | 0.19 | 0.99 |
|  | DIFFAI+ | 0.02 | 1.23 | 0.19 | 0.80 |

Figure 11: Training and Test Results of Thermostat

| Data Size | Approach | $Q$ | $C^{\#}$ | Test Data Loss | Provably Safe Portion |
|---|---|---|---|---|---|
| 150 | DSE | 0.60 | 0.0 | 0.85 | 0.99 |
|  | DIFFAI+ | 0.53 | 21.46 | 0.87 | 0.22 |
| 750 | DSE | 0.64 | 0.0 | 0.86 | 1.0 |
|  | DIFFAI+ | 0.57 | 20.66 | 0.87 | 0.26 |
| 1500 | DSE | 0.58 | 0.0 | 0.84 | 1.0 |
|  | DIFFAI+ | 0.53 | 28.28 | 0.87 | 0.17 |
| 3750 | DSE | 0.62 | 0.0 | 0.85 | 1.0 |
|  | DIFFAI+ | 0.53 | 26.31 | 0.87 | 0.21 |
| 7500 | DSE | 0.61 | 0.0 | 0.86 | 1.0 |
|  | DIFFAI+ | 0.52 | 27.23 | 0.88 | 0.27 |

Figure 12: Training and Test Results of AC

| Data Size | Approach | $Q$ | $C^{\#}$ | Test Data Loss | Provably Safe Portion |
|---|---|---|---|---|---|
| 200 | DSE | 0.22 | 1.12 | 0.28 | 0.66 |
|  | DIFFAI+ | 0.11 | 2.04 | 0.31 | 0.0 |
| 1000 | DSE | 0.22 | 0.0 | 0.25 | 0.99 |
|  | DIFFAI+ | 0.17 | 2.03 | 0.26 | 0.0 |
| 2000 | DSE | 0.21 | 0.0 | 0.25 | 0.99 |
|  | DIFFAI+ | 0.17 | 2.03 | 0.25 | 0.0 |
| 5000 | DSE | 0.22 | 0.0 | 0.27 | 1.0 |
|  | DIFFAI+ | 0.17 | 2.03 | 0.24 | 0.0 |
| 10000 | DSE | 0.22 | 0.0 | 0.24 | 0.99 |
|  | DIFFAI+ | 0.13 | 2.26 | 0.24 | 0.0 |

Figure 13: Training and Test Results of Racetrack

A.9 ADDITIONAL PATTERN ANALYSIS

Figure 2 exhibits safety performance of learnt patterns. The details of the patterns are in Figure 8. We highlight the patterns that cover the scope differences between DIFFAI+ and DSE, including parameterized branch conditions(Pattern 1-4) and bad joins(Pattern 5). In addition, we describe the characteristics in each pattern that influence DSE's performance, including deep local minimum in branches and limitations on sampling. In implementation, we randomly initialize the neural network's parameter (DIFFAI+ and DSE start from the initialization). We observe that all the neural networks are initialized to come with an output $y \leq 1$ when $x \in [-5, 5]$, which gives $0.0$ provably unsafe portion for Pattern 1, 2 and 5 in initialization.

We give a detailed description of each patterns' characteristic (with results analysis), where all the patterns' safety loss is over the only variable $z$ in assertion:

- Pattern1: The output, $y$, of the neural network module is only used as the parameters to calculate the branch condition satisfaction. In each branch, $z$ is only updated with constants. When the gradient back-propagation executes in DIFFAI+, the gradients over $\theta$ is constantly zero since the branch splitting computation is not involved in safety loss. While for DSE, we consider the $y$ by using the volume-based probability in safety loss calculation. Therefore, DSE is able to guide the program to fall into the second branch thoroughly by reducing the probability of $y$ falling into the first branch.

- Pattern2: Similar to Pattern 1, the output of the neural network is used in branch condition. Different from Pattern 1, the input of the neural network, $x$, participates the assignment for $z$. DIFFAI+ has the same "0 gradient" issue as in Pattern 1. DSE sometimes reaches an area during training where only a super small portion of $y$ falls into the first branch and the sampling operation does not pick the portion because the probability is super small. In this way, DSE still steadily converges with a $C^\# = 0.0$. However, the program is safer but not worst-case safe because $C^\# = 0.0$ represents all the trajectories sampled by DSE during training is safe.

- Pattern3: Pattern 3 uses $y$ in the branch condition as well. For the two branches, $z$ in the first branch is directly assigned by a function over $y$ and the second branch is naturally safe. Therefore, the goal for learning safe programs is to optimize $z := 10 - y$ to the safe area. Both DIFFAI+ and DSE success on this pattern only optimizing over one branch is enough to learn a safe program.

- Pattern4: Pattern 4 is a typical example of local minimum. As the branch condition is $y \leq -1.0$, the neural network may be initialized to fall into both branches. The first branch is naturally safe and the second branch comes with a local minimum $z = 2$. DIFFAI+ is not able to learn the safe programs guided by the safe loss and DSE can learn. DSE not achieving $1.0$ provably safe portion as there is a part of training gets stuck in the local minimum with the second branch. There is an exception for DIFFAI+: it gets $0.58$ for NNBig. The $0.58$ can not be viewed as a result benefiting from the safety loss calculation as it specifically benefits from the random initialization. Since this is not a complicated task, DiffAI+ benefits from weights initialization when using NNBig. Specifically, some random initializations of NNBig may pick neural network parameters that make all the $y$ fall into the first branch(safe). This leads to a not bad result of DiffAI+ for Pattern4 with NNBig. If there are other random initializations that can benefit the approach, DiffAI+ and DSE should benefit in the same way. DSE does not specifically benefit from the random initialization for Pattern 4 since DiffAI+ fails on NNSmall and NNMed while DSE does not and DSE gives a safer program on NNBig.

- Pattern5: The first branch in this pattern is a function over the output of the neural network and the second branch is unsafe. There are bad joins for DIFFAI+. For here, one symbolic representation of $y$ falling into both branches always joins with the unsafe results ($z := -10$). There is no gradient from $z := -10.0$ can direct the neural network to the safe area. DSE can work on this pattern as there is a volume-based probability used in learning to guide the $y$ to fall into the first branch.

## A.10 Additional Details of the Core Algorithm

The key point of "sampling based on volume" is the volume-based probability, which is measured by the volume portion satisfying one condition. Therefore, the volume portion must be differentiable (w.r.t $\theta$). In addition, the volume is not required to be non-zero. We give a detailed description as below: - We represent the $V_\theta$ by the box domain (The detailed definition of $V_\theta$ is in Appendix A.2). - When the branch condition splits the box domain into two polyhedra (direct volume calculation for polyhedra is -P hard [1]), we add one new dimension (with a new variable) to allow that the splitting is over this new dimension. Formally, starting from one state $s = (l, V)$, a condition $\text{if} f(x_1, \ldots, x_k) <= M : \ldots$, DSE transforms the program first by converting the condition to $x_{k+1} = f(x_1, \ldots, x_k); \text{if} x_{k+1} <= M : \ldots$. Then, the volume portion is measured by the intersection portion of the concrete interval representation of $x_{k+1}$. Specifically, if the additional variable's concrete interval length is 0, it indicates that the variable represents one point. It falls into one branch (Say branch 1) completely. Then the probability to select branch 1 is 1.0, and the probability to select other branches is 0.0.

## A.11 Additional Scalability Evaluation

For most of the additional experiments for scalability, we take Thermostat. We take AC as the base benchmark for 'different neural network architectures' as we add convolutional layers. There is one variable as the input for the NN in Thermostat, and using Conv1d with kernel size=1 is the same as fully connected layers.

In the original Thermostat, there is a 20-length loop with a dynamic program length of $20 * 6 = 120$. In each iteration, there are 2 branches. Branches stacking on each other would increase the number of branches exponentially. Thus, in the original Thermostat, there is $2^{20}$ paths to go in total. We evaluate the scalability over Thermostat by increase the number of branch in each iteration, doubling the loop length, and use a super refined input size.

### A.11.1 Different Number of Branches

| Data Size | Approach | $Q$ | $C^{\#}$ | Test Data Loss | Provably Safe Portion |
|-----------|----------|-----|----------|----------------|----------------------|
| 200 | Ablation | 0.03 | - | 0.2 | 0.66 |
| | DIFFAI+ | 0.07 | 1.08 | 0.19 | 0.66 |
| | DSE | 0.07 | 0 | 0.25 | 0.99 |
| 10000 | Ablation | 0.01 | - | 0.2 | 0.68 |
| | DIFFAI+ | 0.05 | 1.53 | 0.18 | 0.67 |
| | DSE | 0.06 | 0 | 0.22 | 0.99 |

Figure 14: Results from Thermostat with Three Branches in Each Iteration.

With Thermostat, we increase the number of branches in the iteration from 2 to 3, by allowing the $isOn$ has three branches to go. We convert the branches from {cool, heat} to {cool, lowHeat, highHeat}. That said, the entire number of branches in the program increases from $2^{20}$ to $3^{20}$. We show that DSE still performs well. And DiffAI+ can not give very good results even if ablation gives a reasonable number.

### A.11.2 Different Dynamic Length

With the same structure as the original thermostat, we set the loop length of 40, which doubles the dynamic length and the number of branches increases from $2^{20}$ to $2^{40}$ accordingly.

We show that with 200 data, DSE gives a result better than baselines but definitely less-than-perfect. In the training where DSE did not give safe programs in the cases where the symbolic state (split by too many branches) is too refined. The symbolic state gets stuck in an area where it's not safe but it falls into one trajectory fully. In this way, the volume-based probability is always 1.0 for the later transitions and the gradient over $\theta$ is 0.0. The highly unsafe stuck state gives very high state safety loss, which is exhibited by the large training safety loss for DSE. When DSE is not stuck, it can learn safe programs. That's why the average provable safe portion is still larger than baselines.

| Data Size | Approach | $Q$ | $C^{\#}$ | Test Data Loss | Provably Safe Portion |
|---|---|---|---|---|---|
| 200 | Ablation | 0.03 | - | 0.19 | 0 |
| | DIFFAI+ | 0.05 | 2.05 | 0.21 | 0 |
| | DSE | 0.17 | 396 | 0.26 | 0.36 |
| 10000 | Ablation | 0.01 | - | 0.19 | 0.67 |
| | DIFFAI+ | 0.05 | 25.42 | 0.21 | 0 |
| | DSE | 0.08 | 0 | 0.24 | 0.99 |

Figure 15: Results from Thermostat with 40-Length Loop.

We also show that, with more data, DSE can give a very good safe program for this challenging task while baselines can not. Specifically, DIFFAI+ (splitting 100) easily gets stuck when guided by the highly non-differentiable representation of the safety loss even if ablation gives a reasonable result.

### A.11.3 SUPER REFINED INPUT SIZE

| Data Size | Approach | $Q$ | $C^{\#}$ | Test Data Loss | Provably Safe Portion |
|---|---|---|---|---|---|
| 200 | Ablation | 0.02 | - | 0.19 | 0.33 |
| | DIFFAI+ | 0.04 | 4.5 | 0.18 | 0.33 |
| | DSE | 0.07 | 129.5 | 0.25 | 0.33 |
| 10000 | Ablation | 0.01 | - | 0.19 | 1 |
| | DIFFAI+ | 0.01 | 0 | 0.19 | 1 |
| | DSE | 0.01 | 0 | 0.19 | 1 |

Figure 16: Results from Thermostat with Super Refined Input Size.

With the Thermostat, this task starts from a super refined input size: [60.0, 60.1]. The input in the original Thermostat is [60.0, 64.0]. We can see here DSE does not give good results with 200 data and gives the same result with the Ablation when using the full dataset.

This benchmark exhibits the trade-off between refinement and the ease of learning of DSE. When the input size is small enough that it only fully falls into one trajectory with an updated $\theta$ during learning. In this way, the volume-base probability is also 1.0 for this trajectory and there is not gradient guiding the state to jump from this trajectory to another. Thus, when this case occurs in the training, DSE gets stuck.

### A.11.4 DIFFERENT NEURAL NETWORK ARCHITECTURE

| Data Size | Approach | $Q$ | $C^{\#}$ | Test Data Loss | Provably Safe Portion |
|---|---|---|---|---|---|
| 200 | Ablation | 0.02 | - | 0.19 | 0.33 |
| | DIFFAI+ | 0.04 | 4.5 | 0.18 | 0.33 |
| | DSE | 0.07 | 129.5 | 0.25 | 0.33 |
| 10000 | Ablation | 0.01 | - | 0.19 | 1 |
| | DIFFAI+ | 0.01 | 0 | 0.19 | 1 |
| | DSE | 0.01 | 0 | 0.19 | 1 |

Figure 17: Results from AC with Convolutional Layers

With AC, we use a NN with Conv1d(1, 1, 2)-ReLU()-Conv1d(1, 1, 2)-ReLU()-Linear(4, 32)-ReLU()-Linear(32, 6)-Sigmoid(), where Conv1d(X, Y, Z) means an input channel of X, an output channel of Y and a kernel size of Z and Linear(X, Y) means an input channel of X and an output channel of Y.

The above table exhibits that DSE can work for neural networks with convolutional layers. For AC specifically, all the training data comes from safe trajectories. There is a lack of generality for Ablation to get good test data loss. When adding additional safety constraint(e.g. In DSE), it helps to overcome some local minimum areas and gives a better test data loss even if the training data loss is worse than Ablation.

### A.11.5 CART-POLE EXPERIMENTS

In the Cartpole experiment, we use the trajectories data generated from the expert model from Imitation Package. Starting from $S_0 = [-0.05, 0.05]^4$, we follow Bastani et al. (2018) to train a cart of which the pole keeps upright ($\theta \in [-12 * 2 * math.pi/360, 12 * 2 * math.pi/360]$, the definition of upright is the standard one from OpenAI Gym). The state space is 4 and the action space is 2. We use a 3 layer fully connected neural network (each layer followed by a ReLU and the last layer followed by a Sigmoid) to train. We follow the two points in Bastani et al. (2018) below to setup the experiment:

- We approximate the system using a finite time horizon $T_{max} = 10$;
- We use a linear approximation f(s, a) $\simeq$ As + Ba.

We did the verification by splitting the input space into $20^4$ boxes. And we extract the percentage of safe concrete trajectories from 100 uniformly sampled starting points. The results for restricting pole angle are as shown in Figure 18.

| Approach | $Q$ | $C^\#$ | Test Data Loss | Provably Safe Portion | Percentage of Safe Concrete Trajectories |
|---|---|---|---|---|---|
| Ablation | 0.13 | - | 0.14 | 0.67 | 100% |
| DIFFAI+ | 0.13 | 0.66 | 0.14 | 0.62 | 100% |
| DSE | 0.16 | 0 | 0.18 | 0.78 | 100% |

Figure 18: Results of Cart-Pole(angle)

As indicated by the above figure, pure imitation learning (Ablation) can already learn a pole keeping upright. To show DSE 's ability to train a safe program without too much help from the data loss, we add another experiment following the same setting above except the constraint, which is shown in the main paper.

### A.11.6 TRAIN DSE FROM HIGHLY UNSAFE DATA

| Benchmark | Data Size | Approach | $Q$ | $C^\#$ | Test Data Loss | Provably Safe Portion |
|---|---|---|---|---|---|---|
| Thermostat(45% Unsafety) | 200 | Ablation | 0.04 | - | 0.19 | 0.20 |
| | | DIFFAI+ | 0.06 | 4.40 | 0.21 | 0.20 |
| | | DSE | 0.16 | 0.00 | 0.27 | 0.99 |
| | 10000 | Ablation | 0.04 | - | 0.18 | 0.59 |
| | | DIFFAI+ | 0.05 | 3.72 | 0.17 | 0.46 |
| | | DSE | 0.06 | 0.00 | 0.18 | 0.99 |
| AC(49% Unsafety) | 150 | Ablation | 0.89 | - | 1.04 | 0.32 |
| | | DIFFAI+ | 0.90 | 25.20 | 1.04 | 0.24 |
| | | DSE | 0.96 | 0.00 | 1.03 | 1.00 |
| | 7500 | Ablation | 0.79 | - | 1.03 | 0.00 |
| | | DIFFAI+ | 0.81 | 15.00 | 1.03 | 0.00 |
| | | DSE | 0.88 | 0.00 | 1.03 | 1.00 |

Figure 19: Results from Highly Unsafe Data

In summary, DSE can scale to programs with reasonable length, branches and different neural network architectures. One challenge in DSE is that learning becomes harder when symbolic state representation becomes more refined(including super refined input space, safety constraint, and branch splitting). We leave this open for future works to seek to identify better tradeoffs between precision of symbolic states and ease of learning.

