# OpenReview forum: "Safe Neurosymbolic Learning with Differentiable Symbolic Execution"
_ICLR.cc/2022/Conference — ICLR 2022 Poster_

### Official Review · Reviewer_yj7y · 2021-10-30

**Correctness:** 3
**Technical Novelty And Significance:** 4
**Empirical Novelty And Significance:** 3
**Recommendation:** 6
**Confidence:** 4

**Main Review:**

This paper is well-motivated and targets an important, forward-looking problem: how to learn neurosymbolic programs that are worst-case-safe. The proposed technique of combining symbolic execution with sampling  and REINFORCE-styled learning is interesting. And the experiments show that the proposed method is effective on the benchmarks considered in the paper. I have a few questions regarding the method, which I put below.

Strength
- This paper is well-motivated, targeting an important, forward-looking problem
- The proposed technique is interesting
- Experiments show that the method works well on the benchmarks considered, and also significantly outperforms a prior approach

Weakness/questions
- How are the symbolic states $\sigma$ generated, in particular, the logical formula V’s? Are they human-specified? Seems that they are the key to reduce the trajectory space to a finite set of symbolic trajectories, but I did not find how they are generated in the paper.
- It seems that the proposed REINFORCE-based algorithm essentially works by pushing the parameters of the neural models so that they will only trigger the safe symbolic trajectories. If that is the case, the following simpler method seems to work as well: starting with the set of symbolic trajectories, first determine the subset that are safe, convert the subset into constraints on input/output of the neural model, then train the model directly to satisfy this constraint (by penalization). I would like to see the authors compare to this baseline, or discuss why this would not work.
- In page 3, it was mentioned that the proposed method assumes that the program contains only one learnable neural network. Why does the method require this constraint? This seems to limit the applicability of the proposed method.


**Summary Of The Paper:**

This paper targets the problem of learning parameters of programs that involve both neural and symbolic components, with the objective that the program is guaranteed to be safe. This objective is challenging because it is not differentiable. The proposed method uses symbolic execution to finitize the execution trajectories and uses sampling with REINFORCE-styled algorithm to optimize for an approximated objective. Results show that although the proposed method is not guaranteed to be sound, it in practice produces programs that are safe on the example benchmarks.


**Summary Of The Review:**

This paper is well-motivated and targets an important problem. The proposed method is novel and interesting. However, I have some questions regarding the justification of the proposed method and the limitations on its applicability. Overall, I am slightly leaning towards acceptance, but would like the authors to address my questions before I can solidify my recommendation.

---

> ### Author Response · Authors · 2021-11-17
> **Response to Reviewer yj7y**
>
> Thank you for the feedback! We give the responses as below:
>
> >Q17:
> >How are the symbolic states σ generated, in particular, the logical formula V’s? Are they human-specified? Seems that they are the key to reduce the trajectory space to a finite set of symbolic trajectories, but I did not find how they are generated in the paper.
>
> We represent the logical formula V with the box domain, where each variable is represented by an interval in our implementation (Appendix A.2 gives a detailed definition and computation of the box domain). This is a well-defined symbolic state representation when doing program analysis ([1] has one example on page 243). There are related works using this domain to verify neural networks or learn robust neural networks [2, 3] (using this domain through programs without requiring differentiability in static verification or propagating this domain through isolated neural networks during learning).
>
> >Q18:
> >It seems that the proposed REINFORCE-based algorithm essentially works by pushing the parameters of the neural models so that they will only trigger the safe symbolic trajectories. If that is the case, the following simpler method seems to work as well: starting with the set of symbolic trajectories, first determine the subset that are safe, convert the subset into constraints on input/output of the neural model, then train the model directly to satisfy this constraint (by penalization). I would like to see the authors compare to this baseline, or discuss why this would not work.
>
> Thank you for the proposed method. We believe there are two issues with the simpler method you propose:
> - Given one arbitrary parameter of a neurosymbolic program, we cannot know the set of symbolic trajectories directly without propagating the input space through the program. Such propagation can be very expensive, as a program can have an exponential number of paths (symbolic trajectories). In general, this exponential blowup happens when a program has branches stacked on top of each other. The loops in our benchmarks stack the branches in each iteration. For example, in our benchmarks, covering all the symbolic trajectories can cost $4^{15}$ paths for AC, $2^{20}$ paths for Thermostat and $9^{20}$ paths for Racetrack.
> - Converting safe symbolic trajectories into constraints over input-output of neural networks (or even not requiring differentiability) is non-trivial. Breaking the safe symbolic trajectories into input-output constraints of single neural networks can not cover all the safety cases. Also, many properties of a neurosymbolic program depend on modules correctly interacting with each other and we need to consider the entire program rather than the isolated neural network. Consider the Racetrack benchmark, there are two cars (two neural network planners) on the map. For one planner neural network, the input is the current state’s position and the output is the direction for the next step.  Only when one car’s input-output is fixed can another car’s input-output be defined as safe or unsafe. For example, when car A is in position 1 and it goes down, and the trajectory of it is defined as unsafe because car B below it goes up. We can not say starting from position 1, going down is unsafe for car A because it can also be safe if car B is not below it or car B is below while going down.
>
> >Q19:
> >In page 3, it was mentioned that the proposed method assumes that the program contains only one learnable neural network. Why does the method require this constraint? This seems to limit the applicability of the proposed method.
>
> Thank you for raising this concern. We do not have this limitation. We assumed one learnable network in the technical section only for the brevity of exposition. As mentioned in the footnote of page 3, the method itself supports k (k>1) learnable neural networks in the program. In fact, in our case studies, we learn two neural networks together in both the thermostat($\pi_\theta^\text{cool}$, $\pi_\theta^\text{heat}$) and racetrack($\pi_\theta^\text{agent1}, \pi_\theta^\text{agent2}$) benchmarks.
>
> [1] Cousot, Patrick, and Radhia Cousot. "Abstract interpretation: a unified lattice model for static analysis of programs by construction or approximation of fixpoints." Proceedings of the 4th ACM SIGACT-SIGPLAN symposium on Principles of programming languages. 1977.
>
> [2] Gehr, Timon, et al. "Ai2: Safety and robustness certification of neural networks with abstract interpretation." 2018 IEEE Symposium on Security and Privacy (SP). IEEE, 2018.
>
> [3] Mirman, Matthew, Timon Gehr, and Martin Vechev. "Differentiable abstract interpretation for provably robust neural networks." International Conference on Machine Learning. PMLR, 2018.
>
> [4] Christakis, Maria, et al. "Automated Safety Verification of Programs Invoking Neural Networks." International Conference on Computer Aided Verification. Springer, Cham, 2021.

---

> > ### Comment · Reviewer_yj7y · 2021-11-23
> > **Re: Response**
> >
> > Thank you for the response! The symbolic domain is more clear now.
> >
> > Regarding the exponential number of symbolic trajectories, my understanding is that the proposed method handles it by using sampling. But since it is an exponential number, sampling would not cover all of them either. Do you have a justification on why this is not a problem given that the goal is to train verifiably safe neurosymbolic programs (since now some trajectories are not visited at all)?
> >
> > For the limitation of only one network has trainable parameters, I did see that some case studies have more than 1 trainable networks. But it says in the paper that this limitation is "to ensure that the parameters of the different neural networks in Fθ are not incorrectly entangled". What does this refer to, and how does this "entanglement" affect the proposed method?

---

> > > ### Author Response · Authors · 2021-11-24
> > > **Thank you for the additional response! Here is our response.**
> > >
> > > >Regarding the exponential number of symbolic trajectories, my understanding is that the proposed method handles it by using sampling. But since it is an exponential number, sampling would not cover all of them either. Do you have a justification on why this is not a problem given that the goal is to train verifiably safe neurosymbolic programs (since now some trajectories are not visited at all)?
> > >
> > > We did not find sampling to pose a serious challenge in either our original experiments or the additional experiments during rebuttal. We believe that the reason for this is as follows. Intuitively, in each iteration of stochastic gradient descent, we sample different subsets of paths (symbolic trajectories). Thus, while we do not visit *every* path during every iteration, in aggregate, the training process is likely to receive gradient updates (for the safety loss) from all paths with reasonably high volume, in regular intervals. In addition, note that each path that is sampled is handled soundly. So for each such path, the gradient of the safety loss pushes the learning algorithm towards provable safety for a subregion of the input space.
> > >
> > > It is true that paths with very low volume may not be sampled at all during training, so our stochastic approximation strategy may not work for programs with pathological unsafe trajectories. We believe that in real-world control applications, such pathological trajectories are not commonplace. However, we also intend to pursue orthogonal variance-reduction techniques that can help with handling such anomalous paths in future work.
> > >
> > > > For the limitation of only one network has trainable parameters, I did see that some case studies have more than 1 trainable networks. But it says in the paper that this limitation is "to ensure that the parameters of the different neural networks in Fθ are not incorrectly entangled". What does this refer to, and how does this "entanglement" affect the proposed method?
> > >
> > > We agree that this sentence was a bit unclear, and we will revise it in our next draft.
> > >
> > > The point about 'entanglement' only concerns the data loss and not the safety loss.
> > > Specifically, by ‘entanglement’, we refer to scenarios where NN A’s data is not clearly separated from NN B’s data. Naturally, this can lead to learning incorrect data loss for both networks. To avoid this, in our experiments, we treat the data for different neural networks separately. More precisely, we start with a trajectory dataset, where each trajectory includes multiple input-output datapoints (<state, action>) for different NNs. We divide this input-output data into separate datasets, e.g. dataset_A and dataset_ B. In training/test, dataset_A only goes through NN A and dataset_B only through NN B. This avoids ‘entanglement’ while also allowing us to handle multiple NNs in the same program.
> > >
> > > Take the Thermostat’s data as an example, when the input state has the $isOn <= 0.5$, we mark it as the input for $\pi_{\theta}^\texttt{cool}$ rather than $\pi_{\theta}^\texttt{heat}$. For Racetrack, the trajectories are explicitly marked that trajectory $i$ is for agent1, trajectory $j$ is for agent2.

---

> > > > ### Comment · Reviewer_yj7y · 2021-11-25
> > > > **Thank you for your response**
> > > >
> > > > The response from the authors addressed all my questions. This consolidates my recommendation for acceptance of this paper. I have no further questions at this point.

---

> > > > > ### Author Response · Authors · 2021-11-29
> > > > > **Thank you for the feedback**
> > > > >
> > > > > Thank you very much for the feedback. We enjoyed the discussion with you! Appreciate the detailed questions which help us make a better manuscript!

---

### Official Review · Reviewer_ohyh · 2021-11-01

**Correctness:** 3
**Technical Novelty And Significance:** 4
**Empirical Novelty And Significance:** Not applicable
**Recommendation:** 8
**Confidence:** 4

**Main Review:**

## Novelty and Significance

The proposed approach is both novel and significant. There is an emerging field of approaches for developing and verifying such neurosymbolic programs with safety constraints. DSE represents an advance in that literature which, depending on how well the technique scales to other programs, could emerge as an influential approach in this literature.

## Correctness and Clarity

To the best of my knowledge, the approach presented in the paper is technically sound. My main qualm with the correctness of the approach is in the lack of discussion of the implications of sampling based on volume. Specifically:
- Sampling based on volume:
  - The definition of $V_\theta$ near the top of page 4 is relatively loose. Specifically, it seems $V_\theta$ must consist of shapes with differentiable (w.r.t. $\theta$) and non-zero volume. The choice of intersection of intervals seems to be a sufficient choice (it doesn't seem to allow for the full generality, but that's fine), though it does leave open the possibility of zero volume (e.g., if the shape is lower dimensional than the full space [such as a plane](https://math.stackexchange.com/questions/1697067/lebesgue-n-dimensional-measure-of-a-hyperplane)).
  - In addition to clarifying these points (or correcting me if my understanding of this is wrong), the paper should discuss the case of identically zero volume symbolic states, since this would cause the probability to become undefined.
  - It would also be helpful to see some explicit analysis of the volume and sampling based approach, especially as volumes can behave unintuitively in high dimensions. The paragraph in the middle of page 5 ("Note that a low value ... employed during learning") addresses some of these questions, but the evaluation does not show the probabilities along trajectories induced by this volume-based approach.

I am less convinced of the correctness of the empirical evaluation (corresponding to the final contribution in Section 1). This is due to a combination of the both the benchmarks chosen as well as the evaluation of the results.
- The approach is only validated on programs with control flow, on which DiffAI was not evaluated. DiffAI+ is not a published system, so it's hard to reason about what to expect its performance to be. The paper would be significantly stronger if either the authors compared against a system with well-understood behavior on the given task (i.e., some other neurosymbolic network training system with safety constraints) or the authors additionally evaluate DSE in a setting that DiffAI is validated to perform well in (i.e., on programs without branches).
- The approach is only validated on small-scale programs, using a manual heuristic of splitting the input space into 100 subregions. How was this heuristic decided upon? Does the approach still work when not splitting, or splitting at a finer granularity? Does the approach work on longer programs (with more than 20 lines of code) or when using different neural network architectures? Though of course I don't expect the technique to scale perfectly, it would help to understand where the technique breaks down rather than showing only benchmarks on which it gets perfect or near-perfect results.
- There is also little discussion of the actual results. For instance, there is no analysis of what characteristics of the benchmarks allow DSE to achieve perfect of less-than-perfect results. As a minor point here, the paragraph at the bottom of page 7 also claims that DSE is able to successfully learn pattern2 while DiffAI finds pattern4 hard; however, actual gap in results between the two benchmarks is relatively small (0.58 v.s. 0.78) which makes the claims of success and failure feel arbitrary.
- The paper doesn't show C^# for many problems, meaning there is something of a disconnect between the main body of the paper and the results. Figure 9 shows this over the course of training, though for a tiny dataset size. I would be interested in seeing this for all data points (potentially just the final value). I'm also somewhat confused by DSE's behavior on the racetrack and the thermostat. Is DSE's loss conservative? My understanding was that it is not (that C^# > 0 means that the program is not safe) but both the Thermostat and Racetrack show some variance still, implying to me that the program is unsafe.
- The paper also doesn't show Q^# over the course of training. This especially hampers the understanding of the baseline results -- for instance, why is it that Ablation has consistently high test data loss when it's trained with no constraints? Is this just an issue with generalization?


Moreover, the paper (both approach and evaluation) is not sufficiently clear, such that it would likely not be possible to reproduce the results (or even implement the algorithm) based on the description in the paper.
- Clarity of approach:
  - I'm not super familiar with the prior work, but why convexify the program space? Algorithm 1 shows that this is essentially an iterative algorithm that often will repeat prior steps (with a different value for lambda). Please correct me if I'm misunderstanding something about this.
  - Top of page 5: it would be helpful to define volume here, since it's not immediately clear that the formulae describe shapes (on first pass, I thought that this meant something like the count of satisfying assignments).
  - This is a minor point, but it would be great if the entire set of terms, types, and definitions in the formalism were collected in one place (e.g., as a table in the appendix). I found myself continually having to scroll back and forth and search to find definitions.

- Clarity of evaluation:
  - Many details of the evaluation are left out. Specifically these include the details of the "ground-truth program", the specific loss functions (i.e., the definition of Q^# and C^# for each approach), the meaning of the error bars in Figure 3, the training procedure for the neural networks (i.e., learning rate, convergence criteria, etc.), the approach for interleaving the training of the networks, and the actual architecture of the neural networks.
  - It is also very hard to understand the plots in Figure 4 -- my understanding of the symbolic trajectories is that they should encode intervals of state, but the trajectories seem to be individual lines. It also seems like the symoblic trajectories for the Thermostat allow for the possibility of failure, while Figure 3a seems to show that DSE is verifiably correct.


# Response to author response

Thanks to the authors for the very detailed replies. Based on these clarifications and new results, I plan to raise my score to an accept (under the assumption that these clarifications are integrated into the paper).

## Volume calculation

Thanks to the authors for the clarification here. To confirm my understanding, the description in the paper:

> let Vol($V_\theta$) denote the volume of the assignments to X that satisfy $V_\theta$

is incorrect: for instance in Example_volume, the volume of the assignments to X that satisfy $V_\theta$ is zero, because res=0. However, the example shows that the volume calculation is not actually performed over the entire program state X, and instead is only calculated as the fraction of the range of the singular variable in an if condition.

If this is the case, the authors should correct this description for the final version of the paper. Regardless, the clarification has convinced me that the authors have thought this through and that the approach is sound.


> Could you please clarify further what you mean by “volumes can behave unintuitively in high dimensions”? We are not confident that we fully understand the question.

Sorry for including such a vague statement without an example/citation, I should have been much more clear in the original review. I meant the examples along the lines of the majority of the volume being near the surface of high-dimensional shapes (so e.g. if a vector `x` of length 100 had all elements $\in [0, 1]$, then a branch that checked that all elements were less than 0.9 would likely never be sampled) and that the volume of the unit n-ball tends towards zero. I was curious to see the actual volumes computed by the technique, to understand whether the technique was falling prey to any of these degenerate behaviors.

## Scalability

Thanks for these results -- I would still be interested in seeing results where DSE falls apart even with more data (10,000 does not seem like a particularly large dataset by ML standards, especially considering that this data is synthetically generated, so the fact that DSE still works in this domain gives the impression that these are still small-scale problems), but I'm overall satisfied by this presentation which shows the dimensions on which scalability can be a concern and the knobs to turn to address them.

## Additional experimental details and results

Thanks to the authors for the additional details on the programs, Q, C^#, and the additional analysis. This additional data and analysis seems to be sufficient for readers to much more completely understand the experiments performed, their results, and the comparisons between DSE and the baselines.


**Summary Of The Paper:**

The paper proposes DSE, an approach to optimizing parameters of neurosymbolic programs (programs with mixed neural and symbolic components) while satisfying a safety constraint. To accomplish this, DSE defines a *safety loss*, a nonnegative term in the loss function which is nonzero when the function does not satisfy the safety constraint. This safety loss is constructed by symbolically executing the program, executing discrete transitions probabilistically according to a uniform distribution over concrete states. The paper evaluates DSE on a set of small-scale case studies, showing that in programs with discrete control flow, DSE can often train neural networks that soundly lead to safe behavior, with similar quality of learned results as those of a neurosymbolic training approach which does not take into account safety.

**Summary Of The Review:**

Weak reject. Though the approach is novel and significant and seems correct, the clarity of the description of the approach and clarity and correctness of its evaluation are lacking. I would be willing to raise my score if the authors provided a much more clear description of the algorithm and its evaluation such that it would be possible to reproduce the majority of the results based on the paper alone. I would be willing to raise my score further if the authors provided a more thorough evaluation of the scalability of the approach beyond just neural networks of different parameters (i.e., programs of different static and dynamic lengths, more or less branches, different input sizes and splits, and different neural network architectures).

---

> ### Author Response · Authors · 2021-11-17
> **Response to Reviewer ohyh (Part 1/5)**
>
> Thank you for the feedback! We give the responses as below:
>
> >To the best of my knowledge, the approach presented in the paper is technically sound. My main qualm with the correctness of the approach is in the lack of discussion of the implications of sampling based on volume. Specifically:
> >>Q6.1:
> >>
> >>Sampling based on volume:
> The definition of Vθ near the top of page 4 is relatively loose. Specifically, it seems Vθ must consist of shapes with differentiable (w.r.t. θ) and non-zero volume. The choice of intersection of intervals seems to be a sufficient choice (it doesn't seem to allow for the full generality, but that's fine), though it does leave open the possibility of zero volume (e.g., if the shape is lower dimensional than the full space such as a plane).
>
> The key point of “sampling based on volume” is the volume-based probability, which is measured by the volume portion satisfying one condition. Therefore, the volume portion must be differentiable (w.r.t $\theta$). In addition, the volume is not required to be non-zero. We give a detailed description as below:
> - We represent the $V_\theta$ by the box domain (The detailed definition of $V_\theta$ is in Appendix A.2).
> - When the branch condition splits the box domain into two polyhedra (direct volume calculation for polyhedra is #-P hard [1]), we add one new dimension to allow that the splitting is over this new dimension. Formally, starting from one state $s=(l, V)$, a condition $\texttt{if} f(x_1, \dots, x_k) <= M: \dots$, DSE transforms the program first by converting the condition to $x_{k+1} = f(x_1, \dots, x_k); \texttt{if} x_{k+1} <= M: \dots$. Then, the volume portion is measured by the intersection portion of the concrete interval representation of $x_{k+1}$.
>
> We illustrate the details of the volume portion calculation with one concrete example:
> - ```
> Example_volume(x): # x in [-1, 1]
> 	res = 0
> 	y = NN_\theta(x) # from the computation calculation over the box domain, we get abstract state of y in [0, 2] when x in [-1, 1]
> 	if x + y < 0:
> 		res = x + 10
> 	else:
> 		res = x - 5
> ```
> -Before reaching the branch condition “if x + y <0”, we get a symbolic state with $V_\theta$: x, y, res $\in$ [-1, 1] X [0, 2] X [0, 0].
> We transform the branch condition by adding one variable in an additional dimension. Concretely, the “if x + y < 0” is replaced with:
> ```
> z = x + y
> if z < 0:
> ```
> z is $\in$ [-1, 3] and we consider the splitting over z. We get to two states to select from after the conditional operation:
>  - State 1: x, y, res, z $\in$ [-1, 1] X [0, 2] X [0, 0] X [-1, 0)
>  - State 2: x, y, res, z $\in$ [-1, 1] X [0, 2] X [0, 0] X [0, 3]
>
> - The volume portion to select the first state is $\frac{0 - (-1)}{3 - (-1)} = 0.25$
> The volume portion to select the second state is $\frac{3 - 0}{3 - (-1)} = 0.75$.
> We sample the next transition (to the first state or to the second state) based on the probability distribution [0.25, 0.75].
>
> - Specifically, if the additional variable’s concrete interval length is 0, it indicates that the variable represents one point. It falls into one branch (Say branch 1) completely. Then the probability to select branch 1 is 1.0, and the probability to select other branches is 0.0.
>
> In this way, we reduce the volume computation + probability calculation to volume portion calculation (We do not have to explicitly calculate the volume). Given the condition over one new dimension, the volume portion of it falling into one branch equalling to 0 means that the volume portion of the new dimension falling into other branches equalling to 1.
>
> >>Q6.2:
> >>
> >>In addition to clarifying these points (or correcting me if my understanding of this is wrong), the paper should discuss the case of identically zero volume symbolic states, since this would cause the probability to become undefined.
>
> 0 volume portion is allowed in DSE. Please refer to Q6.1 for a detailed volume-portion calculation.
>
> We equal the volume-portion satisfying one branch condition to the probability to select one transition (from one program step to another). Therefore, the calculation of the volume portion does not need to compute the exact volume of one symbolic state.

---

> > ### Author Response · Authors · 2021-11-17
> > **Response to Reviewer ohyh (Part 2/5)**
> >
> > > Q6 (Cont.)
> > >> Q6.3:
> > >>
> > >> It would also be helpful to see some explicit analysis of the volume and sampling based approach, especially as volumes can behave unintuitively in high dimensions. The paragraph in the middle of page 5 ("Note that a low value ... employed during learning") addresses some of these questions, but the evaluation does not show the probabilities along trajectories induced by this volume-based approach.
> >
> > Thank you for this suggestion! Could you please clarify further what you mean by “volumes can behave unintuitively in high dimensions”? We are not confident that we fully understand the question.
> >
> > We give our attempt to respond to this point as below:
> > - Our volume portion calculation can be generalized to high dimensions. That said, for the box domain, this calculation may not be especially accurate. We plan to pursue the combination of DSE and other domains (i.e. zonotopes, polyhedra) in future work.
> > - Regarding the paragraph you mention, we have revised it in the updated version of the paper.
> > - We exhibit the problem above in Pattern 2, where unsafe trajectories with small volume portion may not be sampled. We give a detailed analysis of the patterns in Appendix A.9. The $C^\#=0$ not implying worst-case safe behavior also occurs in Thermostat. In the table in Q10, Thermostat has a training safety loss of 0.0 while a provably safe portion of 0.99 (< 1.0). This exhibits that the training process with $C^\#=0$ misses at least one symbolic trajectory which is not provably safe.
> >
> > >Q7:
> > >The approach is only validated on programs with control flow, on which DiffAI was not evaluated. DiffAI+ is not a published system, so it's hard to reason about what to expect its performance to be. The paper would be significantly stronger if either the authors compared against a system with well-understood behavior on the given task (i.e., some other neurosymbolic network training system with safety constraints) or the authors additionally evaluate DSE in a setting that DiffAI is validated to perform well in (i.e., on programs without branches).
> >
> > As far as we know, there is no neurosymbolic network training system with safety constraints (with published trajectory data set and the code surrounding the neural network). We are the first attempt to solve this problem. This is also why we generate the data by ourselves.
> >
> > DiffAI does not have general meet and join, only has them for ReLUs.  Therefore, DiffAI can not run (compile) on our benchmarks. DiffAI+ is our attempt at making DiffAI run on our benchmarks. The DiffAI benchmarks (all the benchmarks in DiffAI are isolated neural networks without codes), on the other hand, do not really capture the problem we are trying to solve, which is to handle combinations of NNs and nondifferentiable codes.
> >
> > We tried DSE and DiffAI on straight-line programs (either isolated neural networks or neural networks in single path programs). We observed that the results are almost the same. We give the explanations: for the programs with only one branch or isolated neural networks, DSE is easily reduced to exactly the same solution of DiffAI by sampling only one path (There is only one path in the program), and the path probability is set to 1.0. Forcing DSE to sample K paths is the same because these are duplicated paths and both the probability and the safety loss of each path are the same. Technically, DSE and DiffAI are the same on programs without branches.

---

> > > ### Author Response · Authors · 2021-11-17
> > > **Response to Reviewer ohyh (Part 3/5)**
> > >
> > > > Q8:
> > > > The approach is only validated on small-scale programs, using a manual heuristic of splitting the input space into 100 subregions. How was this heuristic decided upon? Does the approach still work when not splitting, or splitting at a finer granularity? Does the approach work on longer programs (with more than 20 lines of code) or when using different neural network architectures? Though of course I don't expect the technique to scale perfectly, it would help to understand where the technique breaks down rather than showing only benchmarks on which it gets perfect or near-perfect results.
> > >
> > > In our actual experiments involving DSE, we did not split the input space for training. As in the trajectory selection process, we use symbolic execution (as part of DSE), which naturally splits the variable space once encountering the branch if only part of the space satisfies one trajectory.
> > >
> > > We use splitting in DiffAI+ so that it has to perform fewer "bad" joins. This is done specifically to make DiffAI+ stronger and be fair to DiffAI+. **We give a more detailed introduction to DiffAI+ in Appendix A.8.** We also give the additional experimental results of DiffAI+ without splitting input space to support the explanations:
> > >
> > > | Benchmark  | Data Size | Method                  | Test Data Loss | Provably Safe Portion |
> > > |------------|-----------|-------------------------|----------------|-----------------------|
> > > | Thermostat | 200       | DiffAI+ (100 splitting) | 0.21           | 0.28                  |
> > > |            |           | DiffAI+ (no splitting)  | 0.17           | 0                     |
> > > |            | 10000     | DiffAI+ (100 splitting) | 0.19           | 0.8                   |
> > > |            |           | DiffAI+ (no splitting)  | 0.18           | 0                     |
> > > | AC         | 150       | DiffAI+ (100 splitting) | 0.87           | 0.22                  |
> > > |            |           | DiffAI+ (no splitting)  | 0.87           | 0.2                   |
> > > |            | 7500      | DiffAI+ (100 splitting) | 0.88           | 0.27                  |
> > > |            |           | DiffAI+ (no splitting)  | 0.88           | 0.2                   |
> > > | Racetrack  | 200       | DiffAI+ (100 splitting) | 0.31           | 0                     |
> > > |            |           | DiffAI+ (no splitting)  | 0.3            | 0                     |
> > > |            | 10000     | DiffAI+ (100 splitting) | 0.24           | 0                     |
> > > |            |           | DiffAI+ (no splitting)  | 0.24           | 0                     |
> > >
> > > We thank the reviewer for the question of scalability and scope. **We are running the experiments for scalability evaluation and will post the results with additional limitation analysis soon.**
> > >
> > > -- update --
> > > Please check 'Additional Evaluation of the Scalability' for experiments and analysis.
> > > --
> > >
> > > >Q9:
> > > > There is also little discussion of the actual results. For instance, there is no analysis of what characteristics of the benchmarks allow DSE to achieve perfect or less-than-perfect results. As a minor point here, the paragraph at the bottom of page 7 also claims that DSE is able to successfully learn pattern2 while DiffAI finds pattern4 hard; however, actual gap in results between the two benchmarks is relatively small (0.58 v.s. 0.78) which makes the claims of success and failure feel arbitrary.
> > >
> > > We highlight the patterns in Figure 8 (Patterns capture larger benchmarks’ characteristics.) which allow DSE to achieve perfect or less-than-perfect results. **We add a more detailed analysis of these patterns in Appendix A.9.**
> > >
> > > For the minor point mentioned above, we state that 0.78 is a good value compared with 0.0. We say that pattern 4 is hard for DiffAI+ because the 0.58 can not be viewed as a result benefiting from the safety loss calculation. We give the following explanations for this specific value:
> > > - Since this is not a complicated task, DiffAI+ benefits from weights initialization when using NNBig. Specifically, some random initializations of NNBig pick neural network parameters that make all the $y$ fall into the first branch(safe). This leads to a not bad average result of DiffAI+ for Pattern4 with NNBig. If there are other random initializations that can benefit the approach, DiffAI+ and DSE should benefit in the same way. We highlight that DSE does not specifically benefit from the random initialization for Pattern 4 since DiffAI+ and DSE have the same initialization and DiffAI+ fails on NNSmall and NNMed while DSE does not and DSE gives a safer program on NNBig.

---

> > > > ### Author Response · Authors · 2021-11-17
> > > > **Response to Reviewer ohyh (Part 4/5)**
> > > >
> > > > >Q10:
> > > > > The paper doesn't show C^# for many problems, meaning there is something of a disconnect between the main body of the paper and the results. Figure 9 shows this over the course of training, though for a tiny dataset size. I would be interested in seeing this for all data points (potentially just the final value). I'm also somewhat confused by DSE's behavior on the racetrack and the thermostat. Is DSE's loss conservative? My understanding was that it is not (that C^# > 0 means that the program is not safe) but both the Thermostat and Racetrack show some variance still, implying to me that the program is unsafe.
> > > >
> > > > We give all the training data loss ($Q$), training safety loss ($C^#$), test data loss, and provably safe portion for Thermostat below. **The result and performance during the course of training for other cases are in Appendix A.6.** The test data and provably safe portion show the concrete value for Figure 3.
> > > >
> > > > | Benchmark  | Data Size | Method  | training loss | training safety loss | test data loss | provably safe portion |
> > > > |------------|-----------|---------|---------------|----------------------|----------------|-----------------------|
> > > > | Thermostat | 200       | DSE     | 0.13          | 0.02                 | 0.24           | 0.99                  |
> > > > |            |           | DiffAI+ | 0.07          | 25.37                | 0.21           | 0.28                  |
> > > > |            | 1000      | DSE     | 0.09          | 0                    | 0.22           | 0.99                  |
> > > > |            |           | DiffAI+ | 0.05          | 1.96                 | 0.14           | 0.55                  |
> > > > |            | 2000      | DSE     | 0.08          | 0                    | 0.21           | 0.99                  |
> > > > |            |           | DiffAI+ | 0.05          | 2.7                  | 0.15           | 0.46                  |
> > > > |            | 5000      | DSE     | 0.07          | 0                    | 0.19           | 0.99                  |
> > > > |            |           | DiffAI+ | 0.04          | 4.58                 | 0.18           | 0.4                   |
> > > > |            | 10000     | DSE     | 0.04          | 0                    | 0.19           | 0.99                  |
> > > > |            |           | DiffAI+ | 0.02          | 1.23                 | 0.19           | 0.8                   |
> > > >
> > > > DSE’s loss is not conservative. $C^\# > 0$ means that the program is not safe. During training, $C^\# <= 0$ means all the sampled trajectories are safe. However, we can not guarantee the trajectories not sampled are safe. During validation, we verify all the symbolic trajectories by abstract interpretation. Therefore, a 0.0 $C^\#$ during training does not guarantee a 1.0 provable safe portion. But a 1.0 provable safe portion means all the symbolic trajectories (covering all the potential trajectories) are safe, which definitely indicates $C^\#$ is 0.0.
> > > >
> > > > Variance in the provable safe portion exists, which means that we did not learn 100% verified safe starting areas. However, a high provably safe portion gives us a quantitative sense of how provably safe a program is. We design the provable safe portion metric in this paper to present a quantitative measure over the provable safety of a program.
> > > >
> > > > >Q11:
> > > > >The paper also doesn't show Q^# over the course of training. This especially hampers the understanding of the baseline results -- for instance, why is it that Ablation has consistently high test data loss when it's trained with no constraints? Is this just an issue with generalization?
> > > >
> > > > Thanks a lot for this comment. **We add the Q over the course of training when varying data size in Appendix A.6.2.** Specifically, Ablation has similar patterns with DiffAI+ as the safety loss fails to guide DiffAI+. DSE sacrifices some data loss for safety guarantee, which is both shown in figures in Appendix A.6.2 and the detailed training data in the Q10 above.
> > > >
> > > > Yes, we think this is because of the issue with generalization. As shown in the training and test data above, the ablation and DiffAI+ have better training data loss. Considering that we added randomness in data generation and trained two neural networks together, the program did have an issue with generalization. Therefore, only the test data loss in Figure 3 does not explicitly reveal this ‘tension’ between safety loss and data loss.

---

> > > > > ### Author Response · Authors · 2021-11-17
> > > > > **Response to Reviewer ohyh (Part 5/5)**
> > > > >
> > > > > >Clarity of approach
> > > > > >
> > > > > >Q12:
> > > > > >I'm not super familiar with the prior work, but why convexify the program space? Algorithm 1 shows that this is essentially an iterative algorithm that often will repeat prior steps (with a different value for lambda). Please correct me if I'm misunderstanding something about this.
> > > > >
> > > > > The convexification is needed to theoretically guarantee the equivalence between the original constrained optimization problem and the sequence of unconstrained optimization problems that we actually solve. This equivalence requires that the domain of $\theta$ (which is $F_\theta$ as $F$ is only represented by $\theta$) be convex. (Chapter 3.2 in [2] and the first paragraph of Chapter 3 in [3]).
> > > > >
> > > > > In practice, we observed that one or two repeat steps can give us convergence for this outer loop (with one or two different valued lambdas). Of course, we can use an arbitrary large lambda in practice, but there we would lose our theoretical guarantee if we do so.
> > > > >
> > > > > >Q13:
> > > > > >Top of page 5: it would be helpful to define volume here, since it's not immediately clear that the formulae describe shapes (on first pass, I thought that this meant something like the count of satisfying assignments).
> > > > >
> > > > > We do not need to explicitly calculate the volume in DSE. Please refer to Q6.1 for further clarification.
> > > > >
> > > > > >Q14:
> > > > > >This is a minor point, but it would be great if the entire set of terms, types, and definitions in the formalism were collected in one place (e.g., as a table in the appendix). I found myself continually having to scroll back and forth and search to find definitions.
> > > > >
> > > > > Thanks for pointing this out! We will add this table to the paper.
> > > > >
> > > > > >Clarity of evaluation:
> > > > > >
> > > > > >Q15:
> > > > > >Many details of the evaluation are left out. Specifically these include the details of the "ground-truth program", the specific loss functions (i.e., the definition of Q^# and C^# for each approach), the meaning of the error bars in Figure 3, the training procedure for the neural networks (i.e., learning rate, convergence criteria, etc.), the approach for interleaving the training of the networks, and the actual architecture of the neural networks.
> > > > >
> > > > > Here are the details:
> > > > > - **We have added details of the “ground-truth program” in Appendix A.7 of the revised paper.** We will also release all the code for this work (including data generation, datasets, all approaches and experiments).
> > > > > - We add details of the specific loss functions, calculation of error bar, training procedure, etc. in **Appendix A.8**. We give the detailed unsafe function in **Appendix A.3**, where we utilize volume portion without calculating the exact volume.
> > > > > - We give the architecture of the neural networks of three cases in Appendix A.4. Both the NN of cases and patterns are feed-forward neural networks with ReLU or sigmoid as activation layers with a different number of layers.
> > > > >
> > > > > >Q16:
> > > > > >It is also very hard to understand the plots in Figure 4 -- my understanding of the symbolic trajectories is that they should encode intervals of state, but the trajectories seem to be individual lines. It also seems like the symoblic trajectories for the Thermostat allow for the possibility of failure, while Figure 3a seems to show that DSE is verifiably correct.
> > > > >
> > > > > Figure 4 illustrates the symbolic trajectories as sequences of rhombuses. A pair of individual lines form one rhombus sequence. For one symbolic trajectory, the property in one step is represented by one rhombus. The top and bottom points represent the upper bound and the lower bound of one interval. The left and right points represent the step one state is in. For example, in step 14, all x-axises of the left and right points of the rhombuses are 14.0 and 15.0 separately. In addition, if the interval is very short, the rhombus acts like one short parallel line in the figure.
> > > > >
> > > > > Please refer to Appendix A.6.3 to find the exact value in Figure 3. Figure 3 gives 0.99 provably safe portions for Thermostat when splitting the input space into 10000 subregions. 0.99 < 1.0 indicates that there is at least one symbolic trajectory that is not provably safe.
> > > > >
> > > > > [1] Dyer, Martin E., and Alan M. Frieze. "On the complexity of computing the volume of a polyhedron." SIAM Journal on Computing 17.5 (1988): 967-974.
> > > > >
> > > > > [2] Agarwal, Alekh, et al. "A reductions approach to fair classification." International Conference on Machine Learning. PMLR, 2018.
> > > > >
> > > > > [3] Le, Hoang, Cameron Voloshin, and Yisong Yue. "Batch policy learning under constraints." International Conference on Machine Learning. PMLR, 2019.

---

> ### Author Response · Authors · 2021-11-20
> **Additional Evaluation of the Scalability (1/2)**
>
> Thank you very much for pointing out the directions for the evaluation of the scalability.
> >Does the approach work on longer programs (with more than 20 lines of code) or when using different neural network architectures? Though of course I don't expect the technique to scale perfectly, it would help to understand where the technique breaks down rather than showing only benchmarks on which it gets perfect or near-perfect results.
>
> >I would be willing to raise my score further if the authors provided a more thorough evaluation of the scalability of the approach beyond just neural networks of different parameters (i.e., programs of different static and dynamic lengths, more or less branches, different input sizes and splits, and different neural network architectures).
>
> For most of the additional experiments for scalability, we take Thermostat. We take AC as the base benchmark for ‘different neural network architectures’ as we add convolutional layers. There is one variable as the input for the NN in Thermostat, and using Conv1d with kernel_size=1 is the same as fully connected layers.
>
> In the original Thermostat, there is a $20$-length loop with a dynamic program length of $20 * 6=120$. In each iteration, there are 2 branches. Branches stacking on each other would increase the number of branches exponentially. Thus, in the original Thermostat, there are $2^{20}$ paths to go in total. We evaluate the scalability over Thermostat by increasing the number of branches in each iteration, doubling the loop length, and using a super-refined(compared to the safety constraint range and branch condition refinement) input size:
>
> - Different branches:
>
> | Data Size | Method   | Q    | C^\# | Test data loss | Provably Safe Portion |
> |----------|----------|------|------|----------------|-----------------------|
> | 200       | Ablation | 0.03 | /    | 0.2            | 0.66                  |
> |           | DiffAI+  | 0.07 | 1.08 | 0.19           | 0.66                  |
> |           | DSE      | 0.07 | 0    | 0.25           | 0.99                  |
> | 10000     | Ablation | 0.01 | /    | 0.2            | 0.68                  |
> |           | DiffAI+  | 0.05 | 1.53 | 0.18           | 0.67                  |
> |           | DSE      | 0.06 | 0    | 0.22           | 0.99                  |
>
> With Thermostat, we increase the number of branches in the iteration from 2 to 3, by allowing the $isOn$ has three branches to go. We convert the branches from \{cool, heat\} to \{cool, lowHeat, highHeat\}. That said, the entire number of branches in the program increases from $2^{20}$ to $3^{20}$. We show that DSE still performs well. And DiffAI+ can not give very good results even if ablation gives a reasonable number.
>
> - Different Dynamic Length:
>
> | Data Size | Method   | Q    | C^\#  | Test data loss | Provably Safe Portion |
> |-----------|----------|------|-------|----------------|-----------------------|
> | 200       | Ablation | 0.03 | /     | 0.19           | 0                     |
> |           | DiffAI+  | 0.05 | 2.05  | 0.21           | 0                     |
> |           | DSE      | 0.17 | 396   | 0.26           | 0.36                  |
> | 10000     | Ablation | 0.01 | /     | 0.19           | 0.67                  |
> |           | DiffAI+  | 0.05 | 25.42 | 0.21           | 0                     |
> |           | DSE      | 0.08 | 0     | 0.24           | 0.99                  |
>
> With the same structure as the original thermostat, we set the loop length of 40, which doubles the dynamic length and the number of branches increases from $2^{20}$ to $2^{40}$ accordingly.
>
> We show that with 200 data, DSE gives a result better than baselines but definitely less-than-perfect. In the training where DSE did not give safe programs in the cases where the symbolic state (split by too many branches) is too refined. The symbolic state gets stuck in an area where it’s not safe but it falls into one trajectory fully. In this way, the volume-based probability is always 1.0 for the later transitions and the gradient over $\theta$ is 0.0. The highly unsafe stuck state gives very high state safety loss, which is exhibited by the large training safety loss for DSE. When DSE is not stuck, it can learn safe programs. That’s why the average provable safe portion is still larger than baselines.
>
> We also show that, with more data, DSE can give a very good safe program for this challenging task while baselines can not. Specifically, DiffAI+ (splitting 100) easily gets stuck when guided by the highly non-differentiable representation of the safety loss even if ablation gives a reasonable result.

---

> > ### Author Response · Authors · 2021-11-20
> > **Additional Evaluation of the Scalability (2/2)**
> >
> > - Super refined input size:
> > | Data Size | Method   | Q    | C^\#  | Test data loss | Provably Safe Portion |
> > |-----------|----------|------|-------|----------------|-----------------------|
> > | 200       | Ablation | 0.02 | /     | 0.19           | 0.33                  |
> > |           | DiffAI+  | 0.04 | 4.5   | 0.18           | 0.33                  |
> > |           | DSE      | 0.07 | 129.5 | 0.25           | 0.33                  |
> > | 10000     | Ablation | 0.01 | /     | 0.19           | 1                     |
> > |           | DiffAI+  | 0.01 | 0     | 0.19           | 1                     |
> > |           | DSE      | 0.01 | 0     | 0.19           | 1                     |
> >
> > With the Thermostat, this task starts from a super refined input size: [60.0, 60.1]. The input in the original Thermostat is [60.0, 64.0]. We can see here DSE does not give good results with 200 data and gives the same result with the Ablation when using the full dataset.
> >
> > This benchmark exhibits the trade-off between refinement and the ease of learning of DSE.  When the input size is small enough that it only fully falls into one trajectory with an updated $\theta$ during learning. In this way, the volume-base probability is also 1.0 for this trajectory and there is no gradient guiding the state to jump from this trajectory to another. Thus, when this case occurs in the training, DSE gets stuck.
> >
> > - Different neural network architecture:
> > | Data Size | Method   | Q    | C^\# | Test data loss | Provably Safe Portion |
> > |-----------|----------|------|------|----------------|-----------------------|
> > | 150       | Ablation | 0.7  | /    | 0.84           | 0                     |
> > |           | DiffAI+  | 0.73 | 0.54 | 0.81           | 0                     |
> > |           | DSE      | 0.73 | 0    | 0.76           | 1                     |
> > | 7500      | Ablation | 0.67 | /    | 0.84           | 0                     |
> > |           | DiffAI+  | 0.68 | 0.54 | 0.82           | 0                     |
> > |           | DSE      | 0.73 | 0    | 0.76           | 1                     |
> >
> > With AC, we use a NN with  Conv1d(1, 1, 2)-ReLU-Conv1d(1, 1, 2)-ReLU-Linear(4, 32)-ReLU-Linear(32, 6)-Sigmoid, where Conv1d(X, Y, Z) means an input channel of X, an output channel of Y and a kernel size of Z and Linear(X, Y) means an input channel of X and an output channel of Y.
> >
> > The above table exhibits that DSE can work for neural networks with convolutional layers. For AC, all the training data comes from safe trajectories. There is a lack of generality for Ablation to get good test data loss. When adding safety constraints (e.g. In DSE), it helps to overcome some local minimum areas and gives a better test data loss even if the training data loss is worse than Ablation.
> >
> > **In summary, DSE can scale to programs with reasonable length, branches, and different neural network architectures. One challenge in DSE is that learning becomes harder when symbolic state representation becomes more refined(including super refined input space, safety constraint, and branch splitting). We leave this open for future works to seek to identify better tradeoffs between the precision of symbolic states and ease of learning.**

---

> ### Author Response · Authors · 2021-11-22
> **Thank you for the updated feedback!**
>
> We thank the reviewer for the additional feedback and clarification!
>
> >If this is the case, the authors should correct this description for the final version of the paper.
>
> Yes, we will add additional descriptions, correct this, and update the context.
>
> >I would still be interested in seeing results where DSE falls apart even with more data (10,000 does not seem like a particularly large dataset by ML standards, especially considering that this data is synthetically generated, so the fact that DSE still works in this domain gives the impression that these are still small-scale problems), but I'm overall satisfied by this presentation which shows the dimensions on which scalability can be a concern and the knobs to turn to address them.
>
> Thanks for raising this question! We will look into this.

---

### Official Review · Reviewer_X2YM · 2021-11-03

**Correctness:** 3
**Technical Novelty And Significance:** 3
**Empirical Novelty And Significance:** 2
**Recommendation:** 6
**Confidence:** 4

**Main Review:**

**Strengths**

The paper is well written and proposes a mathematically rigorous algorithm to compute gradients in the presence of discontinuous conditionals. The formulation looks sound to me. The need to take gradients through symbolic components arises in many other scenarios, so I am excited to see this idea explored beyond the context of this work.


**Weaknesses**

I have several concerns regarding the current evaluation setup.

— In my opinion, all of the tasks (even the thermostat, ac, racetrack) are synthetic. In particular, I am calling them synthetic because the data to train these models are artificially generated. Thus, the lack of a real-world application of this technique is one of my main concerns.

— I am not convinced about generating the training data from a manually written “safe” program. In these safety-critical domains, one main challenge is the tension between achieving good loss and achieving safety. That seems to be missing in this paper (except probably for the racetrack benchmark).

— These benchmarks actually seem like reinforcement learning problems rather than supervised learning problems. It is not clear to me how the current approach would work in a reinforcement learning setting. Moreover, there are several safe RL works [1, 2, 3] that might form as good baselines to compare this approach to. And these safe RL approaches don’t need to take gradients through the environment code.

— Another potential issue with optimizing through symbolic code is the presence of numerous minima. This issue is a much bigger issue than just being able to take the gradient. I guess that the authors didn’t encounter this local minima issue because of the supervised data (which is collected from a safe program). More realistic benchmarks might reveal this problem.

**Other comments**

Why is safety not measured on concrete trajectories during the evaluation? From figure 4, most of the concrete trajectories from the baselines look safe, but the symbolic trajectories are not safe.


**Summary Of The Paper:**

This paper presents an approach to learn worst-case safe parameters for neurosymbolic programs (programs with neural networks + symbolic portions). The key contribution is an algorithm to compute gradients for the worst-case safety loss by symbolically sampling control paths in the programs and using a modification of the Reinforce estimate to approximate the gradients. The approach is compared with DiffAI on several synthetic tasks.

**Summary Of The Review:**

The evaluation is not satisfactory as it is very synthetic and seems to be tailored to benefit the presented approach. Therefore, I lean towards rejecting this paper.

Update after the author's response: I am happy with the additional experiments and increasing my score to a 6.

---

> ### Author Response · Authors · 2021-11-17
> **Response to Reviewer X2YM (Part 1/3)**
>
> Thank you for the feedback! We give the responses as below:
>
> > Q1:
> > In my opinion, all of the tasks (even the thermostat, ac, racetrack) are synthetic. In particular, I am calling them synthetic because the data to train these models are artificially generated. Thus, the lack of a real-world application of this technique is one of my main concerns.
>
> Thank you for raising this legitimate concern. The challenge in our setting is that here, it is not enough to have a trajectory set  (e.g. traces of human drivers). We also need to have the symbolic code that generates the trajectories and encloses the networks whose parameters we are trying to learn. We are not aware of any existing dataset of such neurosymbolic models. This is why we worked with synthetic benchmarks in this first paper on this problem.
> We tried to carefully design these synthetic benchmarks to mimic real-world scenarios that prior work on program synthesis and neurosymbolic program verification [1, 2] referred to. We will publish these benchmarks as a guide for future research in this direction (we give a detailed description of the data generation process and the ground-truth programs in Appendix A.7). We are also very interested in applying DSE to larger, more real-world control tasks through conversations with control experts. However, this is a longer-term goal and outside the scope of the present work.
>
> > Q2:
> > I am not convinced about generating the training data from a manually written “safe” program. In these safety-critical domains, one main challenge is the tension between achieving good loss and achieving safety. That seems to be missing in this paper (except probably for the racetrack benchmark).
>
> DSE does not require the data to come from a manually written “safe” program. For example, the Racetrack benchmark does not come from manually written “safe” programs as we do not set the constraint that two agents should not collide with each other when generating the data.
>
> That said, we have now performed additional experiments with data generated from “unsafe” programs for Thermostat and AC. As we cannot directly manipulate the level of safety of a program, we do so by adding more noise to the neural network’s output collected by the dataset. The noisy output gives a noisy choice when selecting the branches. We add 25% noise in Thermostat by allowing a 25% $\text{isOn}$ set to $1 - \text{isOn}$ (allows the next step to select another branch). For AC, we add 25% noise by allowing 25% incorrectness when collecting ($\text{p0, p1, p2, p3}$). For example, if one state generated safely of ($\text{p0, p1, p2, p3}$) is [1, 0, 0, 0], we allow it to be one of the $\{[0, 1, 0, 0], [0, 0, 1, 0], [0, 0, 0, 1]\}$ with 25% probability. By adding these noises, we get a thermostat dataset with 45% unsafe trajectories and an AC dataset with 49% unsafe trajectories.
>
> Here are the training and testing results when using different portions of these two datasets:
>
> | Benchmark             | Data Size | Method   | training loss | training safety loss | test data loss | provably safe portion |
> |-----------------------|-----------|----------|---------------|----------------------|----------------|-----------------------|
> | Thermostat(45%unsafe) | 200    | Ablation | 0.04          | /                    | 0.19           | 0.20                  |
> |                       |           | DSE      | 0.16          | 0.00                 | 0.27           | 0.99                  |
> |                       |           | DiffAI+  | 0.06          | 4.40                 | 0.21           | 0.20                  |
> |                       | 10000  | Ablation | 0.04          | /                    | 0.18           | 0.59                  |
> |                       |           | DSE      | 0.06          | 0.00                 | 0.18           | 0.99                  |
> |                       |           | DiffAI+  | 0.05          | 3.72                 | 0.17           | 0.46                  |
> | AC(49%unsafe)         | 150    | Ablation | 0.89          | /                    | 1.04           | 0.32                  |
> |                       |           | DSE      | 0.96          | 0.00                 | 1.03           | 1.00                  |
> |                       |           | DiffAI+  | 0.90          | 25.20                | 1.04           | 0.24                  |
> |                       | 7500   | Ablation | 0.79          | /                    | 1.03           | 0.00                  |
> |                       |           | DSE      | 0.88          | 0.00                 | 1.03           | 1.00                  |
> |                       |           | DiffAI+  | 0.81          | 15.00                | 1.03           | 0.00                  |
>
> These results indicate that the ability of DSE to learn probably safer programs is maintained.

---

> > ### Author Response · Authors · 2021-11-17
> > **Response to Reviewer X2YM (Part 2/3)**
> >
> > >Q2 (Cont.):
> >
> > We agree that there is a tension between achieving a good loss and safety. As indicated by the table above, DSE can sometimes have a poorer training loss than DiffAI+ or Ablation. Even though DiffAI+ also targets learning a safer program, the non-differentiability of the programs and the constant update in branches (These issues give either no gradient for $\theta$ or the gradient for $\theta$ being close to 0 in DiffAI+) for calculating $C(\theta)$ in DiffAI+ make the gradient of C matter less. Therefore, most of the training for DiffAI+ is still guided by Q(data loss), which gives us better training data loss. DSE does sacrifice data loss when learning safe programs in training. This information is not explicitly shown in Figure 3 because the test data loss differences between (Ablation, DiffAI+) and DSE are not significant. Looking into the dataset generation (Appendix A.7 and the generation for unsafe dataset above), this results from the issue of generality considering that there is randomness in data generation and components of an entire neurosymbolic program are trained together (meaning multiple neural networks are trained in the same epoch and the data loss aggregates multiple neural networks’ data loss.)
> >
> > > Q3:
> > >These benchmarks actually seem like reinforcement learning problems rather than supervised learning problems. It is not clear to me how the current approach would work in a reinforcement learning setting. Moreover, there are several safe RL works [1, 2, 3] that might form as good baselines to compare this approach to. And these safe RL approaches don’t need to take gradients through the environment code.
> >
> > Our paper focuses on provably safe imitation learning rather than safe RL. Imitation learning [4] is widely used and we think that provably safe imitation learning is an interesting and challenging problem as well.
> >
> > We agree that RL could be an exciting application setting for our methods. However, there are additional challenges that arise in that setting. In particular, in RL, the environment may be stochastic, so the safety loss needs to be propagated through probabilistic transitions. This is challenging for verified learning, as abstract interpretation for probabilistic programs is ill-understood. We would like to pursue this topic in-depth in follow-up work.
> >
> > > Q4:
> > > Another potential issue with optimizing through symbolic code is the presence of numerous minima. This issue is a much bigger issue than just being able to take the gradient. I guess that the authors didn’t encounter this local minima issue because of the supervised data (which is collected from a safe program). More realistic benchmarks might reveal this problem.
> >
> > Yes, optimizing through symbolic code can be a hard problem because of local minima. However, DSE has a side benefit addressing part of this issue by adding a gradient when only 0 gradient is achieved. Consider the case where one variable $x$ is assigned by two different constants ($x=1, x=2$) in two branches when the branch condition is a function over neural network parameter $\theta$, there are two issues:
> > - Non-differentiability: slight change of $\theta$ makes the value of $x$ jump from 1 to 2.
> > - Bad local minima: slight change of $\theta$ does not change the value of $x$ since $x$ is updated by constant.
> >
> > DSE solves the first issue. For the second, DSE’s method benefits from using the gradient of the volume-based probability, which can quantify whether the slight change of $\theta$ makes the input closer to the $x=1$ branch or the other way round.
> >
> > In fact, our paper includes a microbenchmark that exhibits the issue of local minima (Pattern4; Figure 8). Here, only falling into the first branch can have safe results (The minimum value of $z$ in the second branch is 2.).  If a neural network is initialized to fall into the second branch, then it is easily stuck to optimize over $2 + y^2$ rather than jumping to another branch. In Figure 2, we show that DSE can learn and can give safer programs compared to DiffAI+ for pattern4, which is a method for isolated neural networks (not covering this issue of symbolic codes). Since this is not a complicated task, DiffAI+ benefits from weights initialization when using NNBig. Specifically, some random initializations of NNBig may pick neural network parameters that make all the $y$ fall into the first branch(safe). This leads to a reasonable result of DiffAI+ for Pattern4 with NNBig. If there are random initializations that can benefit the approach, DiffAI+ and DSE should benefit in the same way. We believe that DSE does not specifically benefit from the random initialization for Pattern 4 since DiffAI+ fails on NNSmall and NNMed while DSE does not and DSE gives a safer program on NNBig.
> >
> > In addition, as shown in our response to Q2, converting manually written “safe” programs to manually written “unsafe” programs does not harm DSE. Thus, ‘safe’ data is not a trick we used to benefit our approach.

---

> > > ### Author Response · Authors · 2021-11-17
> > > **Response to Reviewer X2YM (Part 3/3)**
> > >
> > > > Q5:
> > > >Why is safety not measured on concrete trajectories during the evaluation? From figure 4, most of the concrete trajectories from the baselines look safe, but the symbolic trajectories are not safe.
> > >
> > > We add the percentage of the safe concrete trajectories here (the order mapping with Figure 4):
> > >
> > > |                     | Ablation(Final) | DiffAI+(Final) | DSE(Middle) | DSE(Final) |
> > > |---------------------|-----------------|----------------|-------------|-------------
> > > | Thermostat          | 0.00%           | 82.80%         | 100%        | 100%        |
> > > | AC                  | 0.00%           | 0.00%          | 100%        | 100%        |
> > > | Racetrack(position) | 90.90%          | 100%           | 100%        | 100%        |
> > > | Racetrack(distance) | 0.00%           | 0.00%          | 100%        | 100%        |
> > >
> > > For Racetrack, safety means satisfying the position and distance constraint at the same time.
> > >
> > > In DiffAI+ for Thermostat and Ablation for Racetrack(position), there is a portion of concrete trajectories that are unsafe.
> > >
> > > We would also emphasize that the goal of this paper is to give a method to learn provably safer neurosymbolic programs. Measuring the safety of concrete trajectories gives readers a sense of the safety of a program. However, the percentage of the safe concrete trajectories can not exhibit the provable safety of a program. In evaluation, we split the input area evenly into 10000 subregions and represent them with box soundly (the input area is a subset of the union of this 10000 subregions) and verify the safety of the symbolic trajectories starting from these 10000 subregions using abstract interpretation [3]. That said, the 10000 symbolic trajectories provably cover all the possibilities of the trajectories starting from the input area. Each symbolic trajectory covers all the possibilities of the trajectories starting from the starting subregion.
> > >
> > > Figure 4 illustrates the symbolic trajectories when we split the input area into 100 subregions in testing (Plotting 10000 subregions split makes symbolic trajectories more refined but the figures are more messy and hard to read. Therefore, we show provably safe portions from 10000 subregions in the text while plotting symbolic trajectories with 100 subregions). Each symbolic trajectory is represented by a sequence of green rhombuses. Although in some cases, most of the concrete trajectories are safe, some of the corner cases not covered by the concrete trajectories (while symbolic trajectories cover all the cases) are not safe (the program is still not a provably safer program).
> > >
> > > [1] Chaudhuri, Swarat, Martin Clochard, and Armando Solar-Lezama. "Bridging boolean and quantitative synthesis using smoothed proof search." Proceedings of the 41st ACM SIGPLAN-SIGACT Symposium on Principles of Programming Languages. 2014.
> > >
> > > [2] Christakis, Maria, et al. "Automated Safety Verification of Programs Invoking Neural Networks." International Conference on Computer Aided Verification. Springer, Cham, 2021.
> > >
> > > [3] Cousot, Patrick, and Radhia Cousot. "Abstract interpretation: a unified lattice model for static analysis of programs by construction or approximation of fixpoints." Proceedings of the 4th ACM SIGACT-SIGPLAN symposium on Principles of programming languages. 1977.
> > >
> > > [4] Hussein, Ahmed, et al. "Imitation learning: A survey of learning methods." ACM Computing Surveys (CSUR) 50.2 (2017): 1-35.

---

> > > > ### Comment · Reviewer_X2YM · 2021-11-20
> > > > **Reply to safe concrete trajectories expt**
> > > >
> > > > I do understand that safety with concrete trajectories does not imply provable safety. Nevertheless, it is a good metric to judge the complexity of the benchmarks chosen in this paper.
> > > >
> > > > Moreover, since the abstract trajectories are computed using the author's proposed abstraction algorithm, it is not clear if the abstract safety is worse because of the abstraction itself or because of the baseline algorithm.

---

> > > > > ### Author Response · Authors · 2021-11-21
> > > > > **Thank you for the additional comments for Q4.**
> > > > >
> > > > > Thanks for pointing this out! Yes, we have over-approximations in the evaluation. Therefore, we showed the concrete trajectories in Figure 4 to show that DSE is indeed better than DiffAI+ and Ablation. Adding the percentage is definitely more clear! We also added the percentage of concrete trajectories in the cart-pole benchmark above. We hope this can address your concern.

---

> > > ### Comment · Reviewer_X2YM · 2021-11-20
> > > **Thank you for the response**
> > >
> > > Thank you for the response and the new results.
> > >
> > > I particularly like the authors' way of characterizing the application domain as safe imitation learning. That said, I am still not convinced about the chosen benchmarks in this paper. I am worried that the current benchmarks and data-generation process are too simple in terms of the state-space, action-space, and the complexity for the NN controller.
> > >
> > > It should be possible for the authors to try the approach on interesting imitation learning tasks (for e.g. OpenAI gym environments or [1] below -- most of their environments can be converted into a symbolic program and it should be possible to get imitation traces using existing RL algorithms).
> > >
> > > [1] Bastani, Osbert, Yewen Pu, and Armando Solar-Lezama. "Verifiable reinforcement learning via policy extraction." arXiv preprint arXiv:1805.08328 (2018).

---

> > > > ### Author Response · Authors · 2021-11-21
> > > > **Additional Experiments with Cart-Pole**
> > > >
> > > > Thank you very much for pointing out the additional benchmarks. We have performed additional experiments on the cartpole benchmark from [1] and summarize the results below. We are currently investigating the remaining benchmarks in the paper.
> > > >
> > > > In the Cartpole experiment, we use the **trajectories data generated from the expert model in [Imitation Package](https://github.com/HumanCompatibleAI/imitation/tree/master/tests/testdata/expert_models/cartpole_0)**. Starting from $S_0=[-0.05, 0.05]^4$, we follow [1] to train a cart of which the pole keeps upright ($\theta$ $\in$ [ -12 * 2 * pi/360, 12 * 2 * pi/360]), the definition of upright is the standard one from OpenAI Gym). The state space is 4 and the action space is 2. We use a 3 layer fully connected neural network (each layer followed by a ReLU and the last layer followed by a Sigmoid) to train. We follow the two points in [1] below to set up the experiment:
> > > > - We approximate the system using a finite time horizon $T_{max} = 10$.
> > > > - We use a linear approximation f(s, a) ≈ As + Ba.
> > > >
> > > > We did the verification by splitting the input space into ${20}^4$ boxes. And we extract the percentage of safe concrete trajectories from 100 uniformly sampled starting points.
> > > > The results for restricting pole angle are as follows:
> > > >
> > > > |          | Q    | C^\# | Test Data Loss | Provably Safe Portion | Percentage of Safe Concrete Trajectories |
> > > > |----------|------|------|----------------|-----------------------|------------------------------------------|
> > > > | Ablation | 0.13 | /    | 0.34           | 0.67                  | 100%                                     |
> > > > | DiffAI+  | 0.13 | 0.66 | 0.34           | 0.62                  | 100%                                     |
> > > > | DSE      | 0.16 | 0    | 0.28           | 0.78                  | 100%                                     |
> > > >
> > > > As indicated by the above figure, pure imitation learning(Ablation) can already learn a pole keeping upright. As the training data is almost all ‘safe’, DSE’s generability(test data loss) benefits from the additional safety constraint.
> > > >
> > > > To show DSE’s ability to train a safe program without too much help from the data loss, we add another experiment following the same setting above except the constraint. We require the cart’s position to be within a range. That said, the cart keeps in the middle of the figure. We set the constraint of cart's position $x$ in the trajectory to be $[-0.1, 0.1]$:
> > > >
> > > > |          | Q    | C^\# | Test Data Loss | Provably Safe Portion | Percentage of Safe Concrete Trajectories |
> > > > |----------|------|------|----------------|-----------------------|------------------------------------------|
> > > > | Ablation | 0.13 | /    | 0.34           | 0.3                   | 41%                                      |
> > > > | DiffAI+  | 0.13 | 0.64 | 0.34           | 0.21                  | 36%                                      |
> > > > | DSE      | 0.22 | 0    | 0.35           | 0.97                  | 100%                                     |
> > > >
> > > > From the above table, we can see that DSE is better than baselines in terms of safety. We also added the concrete trajectories measurement to clarify the safety measurement with provably safe portions.
> > > >
> > > > As a minor point here, we show AC(state space:6, action space:4, neural network node: 192[we can increase the nodes if you think necessary]), Racetrack(state space: 4, action space:6, 2 neural networks both with 192 hidden nodes) in our paper. We think they are comparable with the benchmarks in [1] in terms of state space, action space. We would like to increase the nodes to train if the reviewer feels it helps address the concerns.
> > > >
> > > > In summary, thank you again for pointing out the benchmarks without self-data generation! We think they are all interesting. Please let us know if this response helps address your concern or not. We look forward to receiving your feedback!
> > > >
> > > > [1] Bastani, Osbert, Yewen Pu, and Armando Solar-Lezama. "Verifiable reinforcement learning via policy extraction." arXiv preprint arXiv:1805.08328 (2018).

---

> > > > > ### Comment · Reviewer_X2YM · 2021-11-24
> > > > > **Reply to the authors**
> > > > >
> > > > > Thanks for the new experiments on the cartpole benchmark. The new results certainly increase my confidence in the paper. I am increasing my score to a 6.
> > > > >
> > > > > On a side note, the current appendix of the paper is too long, please carefully figure out what needs to be included in the main paper vs appendix.

---

> > > > > > ### Author Response · Authors · 2021-11-29
> > > > > > **Thank you for the feedback**
> > > > > >
> > > > > > Thank you very much for the insightful discussion and questions! Your suggestions and questions help us make a stronger manuscript.
> > > > > >
> > > > > > Thanks for the side note. We will carefully select the materials to be included in the main paper.

---

> ### Comment · Reviewer_X2YM · 2021-11-17
> **Missing citations**
>
> Sorry about missing the citations. Here they are:
>
> [1] Le, Hoang, Cameron Voloshin, and Yisong Yue. "Batch policy learning under constraints." International Conference on Machine Learning. PMLR, 2019.
>
> [2] Kalweit, Gabriel, et al. "Deep constrained q-learning." arXiv preprint arXiv:2003.09398 (2020).
>
> [3] Achiam, Joshua, et al. "Constrained policy optimization." International Conference on Machine Learning. PMLR, 2017.

---

> > ### Author Response · Authors · 2021-11-18
> > **Thanks for sharing the citations! We give additional responses for Q3.**
> >
> > Thank you for sharing these citations! We give additional responses for Q3:
> >
> > As we have said in Q3 in “Response to Reviewer X2YM (Part 2/3)”, we are focusing here on imitation learning. If one has RL with blackbox environments (the setting in the above paper [1,2,3]), formal verification over this setting is not well-defined. Specifically, verifying over blackbox(e.g APIs) is not feasible. Therefore, we currently do not have a method to evaluate the provable safety. If one has whitebox environments then there are extra challenges (probabilistic environments) as stated in Q3.
> >
> > We can consider restricted properties that only relate to the inputs and outputs of the policy. In this setting, we could imagine doing verification and applying a verified learning approach. But for DSE’s problem (backpropagation through symbolic components) to make sense, the policy representation needs to be neurosymbolic (and potentially discontinuous, patterns in Figure 8 could be simple examples). This would immediately be more general than the setting in [1, 2, 3], in which policies are required to be differentiable.
> >
> > [1] Le, Hoang, Cameron Voloshin, and Yisong Yue. "Batch policy learning under constraints." International Conference on Machine Learning. PMLR, 2019.
> >
> > [2] Kalweit, Gabriel, et al. "Deep constrained q-learning." arXiv preprint arXiv:2003.09398 (2020).
> >
> > [3] Achiam, Joshua, et al. "Constrained policy optimization." International Conference on Machine Learning. PMLR, 2017.

---

### Author Response · Authors · 2021-11-21
**Summary of Our Response**

We would like to thank all the reviewers for their very detailed, helpful, and constructive comments.

We’ve uploaded the response for all reviewers, updated a revised draft incorporating reviewer feedback, and added additional experiments. Below is a summary of the main points:
- (**R1**)We added the experiments from unsafe data and experiments over cart-pole. (Appendix A.11.5, A.11.6)
- (**R1, R2, R3**)We added the core algorithm details, evaluation clarification, and more analysis of the results, as discussed in responses to individual reviews. (Appendix A.6-A.10)
- (**R2**)We added additional experiments for scalability evaluation, including different dynamic lengths, different input sizes, different numbers of branches, and different NN architectures. (Appendix A.11)
- (**R2**)We added additional clarification for the baselines(DiffAI+). (Section 5)
-  (**R2**)We revised the zero-volume description. (Section 4)
- In the revised paper, all the revised text is marked blue.


We hope we have addressed your concerns and look forward to receiving your updated feedback.

---

### Decision · Program_Chairs · 2022-01-20

**Decision:**

Accept (Poster)

**Comment:**

The proposed method, Differentiable Symbolic Execution (DSE), addresses the safety of learned navigation and control programs. The approach samples code paths using a softened probabilistic version of symbolic execution,  constructing gradients of a "safety loss" along these paths, and then backpropagating these gradients through program operations using RL.

Pros
 - The paper is well-written and sound
 - The issue of safety is underexplored
 - The method improves over a strong baseline on benchmarks

Cons
 - The benchmarks are relatively small-scale and artificial